



# Historical Climate and Future Projection in the North Atlantic and Arctic: Insights from EC-Earth3 High-Resolution Simulations

Mehdi Pasha Karami[1], Torben Koenigk[1,2], Shiyu Wang[1], René Navarro Labastida[1], Tim Kruschke[3], Aude Carreric[4], Pablo Ortega[4], Klaus Wyser[1], Ramon Fuentes Franco[1], Agatha M. de Boer[2,5], Marie Sicard[2,5,6], Aitor Aldama Campino[1]

1. Rossby Centre, Swedish Meteorological and Hydrological Institute (SMHI), Norrköping, Sweden
2. Bolin Center for Climate Research, Stockholm, Sweden
3. Federal Maritime and Hydrographic Agency (BSH), Hamburg, Germany
4. Barcelona Supercomputing Center (BSC), Barcelona, Spain
5. Department of Geological Sciences, Stockholm University, Stockholm, Sweden.
6. School of Earth and Environment, University of Leeds, Leeds, UK

*Correspondence to*: Pasha Karami (pasha.karami@smhi.se)

**Abstract**

This study presents a new set of high-resolution global climate simulations conducted with the EC-Earth3 model, including a 350-year pre-industrial, followed by historical (1850–2014) and future (2015–2100, SSP2-4.5) simulations. The model features a horizontal resolution of ~40 km in the atmosphere and 0.25° in the ocean. The high-resolution EC-Earth3 (EC-Earth3-HR) is compared to the standard-resolution version used in CMIP6 to assess the impact of increased resolution on the representation of key climate variables, focusing particularly on the Arctic and North Atlantic regions. The high-resolution model aligns more closely with reanalysis data, particularly for global mean surface temperature and sea surface temperature. Both model resolutions exhibit similar biases in North Atlantic sea surface temperature and salinity, and in Arctic sea ice concentration, although the higher-resolution version shows regional improvements. The EC-Earth3-HR model captures the observed AMOC variability in the early 2000s, along with the trend and rapid loss event in Arctic sea ice. For future projections under SSP2-4.5, the high-resolution model projects a nearly ice-free Arctic by 2040—earlier than the standard-resolution model—while simulating less Arctic warming and a more pronounced weakening of the AMOC. Furthermore, we present a novel method for estimating deep water formation rates and examining the processes contributing to the weakening of the AMOC. Our analysis shows that, in future projections, the Labrador Sea is responsible for the weakening of the AMOC, while the Irminger Basin, which has the strongest contribution to the AMOC, plays a crucial role in sustaining it. In these projections, deep water formation in the Labrador Sea undergoes a complete shutdown, while it decreases by 62% in the Greenland Sea and only 13% in the Irminger Sea.

## 1. Introduction

Anthropogenic emissions of greenhouse gasses and aerosols, as well as human-induced land-use change, are the main drivers of global climate change (IPCC, 2021). The warming is largest in the Arctic, where the temperature is increasing more than three times as fast as the global mean temperature (Rantanen et al. 2022). This warming is closely associated with a strong decline in Arctic sea ice. The annual mean area of Arctic sea-ice has decreased by about 2 million km² since 1980 (IPCC 2021, Onarheim et al. 2018), and the mean sea ice thickness has decreased by about 1.5 to 2 meters since 1980's (Kwok, 2018). Climate models contributing to the sixth phase



of the Coupled Model Intercomparison Project (CMIP6) agree that sea ice will continue to decline in the future under all emission scenarios. However, there is a wide range among the models (Notz and SIMIP Community, 2020) and the timing of a summer ice-free Arctic remains uncertain. A selection of CMIP6 models that are in better agreement with observations suggest that summer sea ice could be completely lost by 2035, and winter sea ice might be rapidly reduced with global mean temperature warming exceeding 3-4°C (Docquier and Koenigk 2021). The weakening of the mid-latitude westerlies and the negative phase of the North Atlantic Oscillation (NAO) were found as a robust response to sea-ice change in winter (Deser et al., 2015; Screen et al., 2013; Wyser et al., 2021). In addition, the global ocean is changing rapidly (IPCC, 2021; IPCC SROC, 2019) and the global mean ocean temperature is currently warmer than at any time observed in the modern era. In spring and summer 2023, the North Atlantic experienced unprecedented ocean temperature anomalies, and the causes and consequences of such events are not yet fully understood.

Large-scale and long-term variations in the temperature of the North Atlantic Ocean are closely linked to the variability of the Atlantic Meridional Overturning Circulation (AMOC). Indirect measurements and reconstructions indicate that the AMOC has been weakening since the mid-20th century (Caesar et al., 2021) and that it is currently at its lowest values since at least 1600 years (Rahmstorf et al., 2015, Tornally et al. 2018). However, the reliability of the reconstructions is debated and opposing results show that the AMOC has not weakened in the last 30 years (Worthington et al., 2021). Moreover, while climate models agree that the Gulf Stream system and the overturning circulation will weaken in the future (Weijer et al., 2019), they largely disagree on the rate of weakening, and the processes behind the simulated weakening. A weakening of the AMOC will affect the spatial distribution of sea surface temperatures, which in turn may affect atmospheric circulation and regional climate (Jackson et al., 2015). In today's climate, Arctic sea ice extent is also negatively correlated with the strength of the AMOC, which is expected to be less pronounced under warmer climate conditions in general (DeBoer et al., 2008).

Results from the High Resolution Model Intercomparison Project (HighResMIP) have highlighted the implications and benefits of increasing model resolution for the representation of key climate processes in the North Atlantic – Arctic climate system (Haarsma et al., 2020). While higher resolution does not necessarily reduce large-scale biases, resolution can be particularly important for modelling the complex topography and small straits of the Arctic (e.g. the Canadian Archipelago and Bering Strait). Increasing the ocean resolution to ~0.25° has been shown to lead to enhanced northward oceanic heat fluxes in the North Atlantic (Grist et al., 2018), which in turn contributes to reduced Arctic sea ice (Docquier et al., 2019). It is evident that high-resolution models offer a more realistic representation of the observed pathways of warm Atlantic water entering the Arctic, particularly through the Fram Strait and the Barents Sea (Docquier et al., 2019). Additionally, the AMOC in higher resolution models tends to be more sensitive to climate change and shows a stronger reduction compared to lower resolution models (Jackson et al., 2020, Roberts et al., 2020). These models also reduce biases in temperature and salinity in the North Atlantic and a more realistic representation of the vertical temperature and salinity profiles in convection regions, but tend to overestimate the deep mixed volume in the convection regions (Koenigk et al., 2021). Furthermore, increasing ocean resolution improves the representation of the position of the Gulf Stream, along with its associated sea surface temperature gradients and



variability (Siqueira and Kirtman, 2016). This, in turn, has been shown to improve the atmospheric circulation by
better capturing the poleward shift of storm tracks and surface heat fluxes (Foussard et al., 2019). Increased
resolution also leads to improved representation of, for example, weather regimes (Fabiano et al., 2020) and
blocking (Schiemann et al., 2017) in the North Atlantic regions, as well as  moisture transport to the continents
(Vanniere et al., 2018) and extreme precipitation (Bador et al., 2020).
Despite the aforementioned improvements enabled by the enhanced resolution, the HighResMIP protocol was
designed to isolate the effect of resolution increase and keep computational resource usage at a reasonable level.
Thus, the tuning of the high-resolution versions was limited to by-definition resolution-dependent parameters,
the spin-up was kept short (30-50 years) and the simulations covered only the period 1950-2050, which might
have prevented the identification of additional improvements. Here, we use a higher-resolution version of
EC-Earth3 global coupled climate model (EC-Earth3-HR) to address some shortcomings of the HighResMIP
protocol. The goal of this study is to present a new set of high-resolution simulations using the most recent, tuned
version of EC-Earth3-HR. It has undergone a tuning process, a spin-up phase, and 350 years of pre-industrial
control simulations. These were followed by a full historical simulation (1850–2014) and a future projection
until 2100 under the SSP2-4.5 scenario. Our primary focus is to analyze key aspects of ocean and sea ice
conditions in the Arctic–North Atlantic region and their projected future changes. Additionally, we introduce a
new method to estimate the rate of deep water formation.
**2.  Model, tuning and simulations**
EC-Earth3, which was used for CMIP6 in its standard resolution (Döscher et al. 2022), serves as the basis for
EC-Earth3-HR and is hereafter referred to as EC-Earth3-SR for easier comparison with EC-Earth3-HR. As in the
standard configuration, EC-Earth3-HR consists of the atmosphere model IFS (cycle 36r4) including the land
surface module HTESSEL and the ocean model NEMO, version 3.6 including the sea ice model LIM3.
EC-Earth3-HR uses a T511 spectral resolution (approx. 40 km) and 91 vertical levels for the atmosphere and
0.25˚ resolution and 75 layers in the ocean, the so-called ORCA025 configuration of NEMO, while
EC-Earth3-SR used T255 and ORCA1 (with grid spacings of  ~80 km and 1°, respectively). An earlier version of
EC-Earth3-HR, known as EC-Earth3P-HR, contributed to HighResMIP (Haarsma et al., 2016) and has been
described in detail by Haarsma et al. (2020). It is also featured in Moreno-Chamarro et al. (2025) alongside its
eddy-resolving counterpart, EC-Earth3P-VHR. Following the HighResMIP-protocol, EC-Earth3P-HR did not
undergo a tuning process, but used the parameter values from the EC-Earth3-SR version to the extent possible.
The version employed here differs from that used in EC-Earth3P-HR in several respects. Firstly, the latter made
use of stratospheric aerosols in a simplified manner, neglecting the indirect aerosol effect. Moreover, vegetation
and its albedo were based on present-day climatological data.
A major difference between EC-Earth3P-HR and EC-Earth3-HR is that EC-Earth3-HR has gone through a
tuning process. While this tuning process in EC-Earth3-HR generally followed the tuning procedure described in
Döscher et al. (2022), due to the high computational costs of running the high-resolution version, the length of
the tuning simulations was shorter and a reduced number of tuning parameters were optimized. First, the
atmosphere was tuned in atmosphere stand-alone (AMIP) runs. Tuning was aimed at minimizing net surface and



top of the atmosphere (TOA) radiative fluxes and limiting the climate drift before reaching radiative equilibrium.
The major challenge was to achieve a stable model state with relatively short tuning runs. To achieve the desired
energy balance and to reproduce reasonably the observed climate, the model was fine-tuned by adjusting some
parameters of sub-grid scale parameterizations, mostly related to cloud and microphysical convection processes,
which effectively control the radiation balance. Compared to the EC-Earth3-SR, different but plausible
parameters were used to achieve the desired radiation balance, which are listed in Table1.

Further tuning was then carried out by focusing on the ocean in coupled pre-industrial experiments. The first
coupled tuning experiment started from a Levitus climatology using the parameter values of the NEMO version
of EC-Earth3-SR. The tuning parameters are similar to those used for the tuning of EC-Earth3-SR (Döscher et
al., 2022) and EC-Earth3P-HR in its second configuration (Haarsma et al., 2020). In our study, we evaluated
various parameters with the aim of improving the model (Table 2). We tested different advection schemes,
including the Upstream-Biased Scheme (UBS), Total Variations Diminishing (TVD) approach, and a TVD
scheme with sub-time stepping of vertical tracer advection (TVD_ZTS). Among these, the TVD scheme was
ultimately selected for its overall better performance. The turbulent kinetic energy (TKE) mixing below the
mixed layer was set to zero (nn_etau=0), as in EC-Earth3-SR (Döscher et al., 2022), which would otherwise lead
to a significant reduction in AMOC. With regard to the horizontal eddy diffusivity of tracers, the lateral mixing
coefficient (rn_aht) was adjusted to a constant value of 1000 $m^2/s$ , in contrast to 300 $m^2/s$ in the EC-Earth3P-HR
configuration. The Langmuir cell coefficients were maintained at 0.2, in accordance with EC-Earth3-SR and
EC-Earth3P-HR. The thermal conductivity of snow (rn_cdsn) was reduced to 0.15 W/m/K, in comparison to 0.27
W/m/K in EC-Earth3-SR and to 0.40 W/m/K in EC-Earth3P-HR, resulting in thinner sea ice and higher surface
skin temperatures. The albedo of sea ice and snow on ice was adjusted, specifically the melting snow albedo on
ice (rn_alb_smlt), to 0.72 (from 0.75). This resulted in earlier melting of snow on ice and reduced ice thickness
in summer. Finally, the freshwater correction value was slightly modified to oas_mb_fluxcorr=1.07945 to
conserve ocean seawater mass; as for EC-Earth3-SR.

**Table1. Atmospheric tuning parameters changed in T511L91  compared to T255L91.**

| IFS parameter | description | T511L91 | T255L91 |
|---|---|---|---|
| RPRCON | Rate of conversion of cloud water to rain | $1.2 \times 10^{-3}$ | $1.34 \times 10^{-3}$ |
| RVICE | Fall speed of ice particles | 0.15 | 0.137 |
| ENTRORG | Entrainment in deep convection | $1.8 \times 10^{-4}$ | $1.7 \times 10^{-4}$ |
| RLCRIT_UPHYS | Critical droplet radius for autoconversion in large-scale precipitation | $0.935 \times 10^{-5}$ | $0.875 \times 10^{-5}$ |
| DETRPRN | Detrainment rate in penetrative convection | $0.85 \times 10^{-4}$ | $0.75 \times 10^{-4}$ |





**Table 2. Ocean parameters tested during tuning and their final values.**

| NEMO Parameter | Description | EC-Earth3-HR | EC-Earth3-SR |
|---|---|---|---|
| Advection Schemes | Different schemes tested; TDV selected for use. | TVD | TVD_ZTS |
| Turbulent Kinetic Energy Mixing | Mixing below the mixed layer (nn_etau) | 0 | 0 |
| Horizontal Eddy Diffusivity | Diffusivity for tracers. (rn_aht_0) | 1000 | 1000 |
| Langmuir Cells Coefficient | Coefficient for Langmuir cells simulation (rn_lc) | 0.2 | 0.2 |
| Thermal Snow Conductivity | Conductivity reduced to affect sea ice thickness and surface temperature (rn_cdsn) | 0.15 | 0.27 |
| Sea Ice and Snow Albedo | Albedo tested, specifically the melting snow albedo on ice (rn_alb_smlt) | 0.72 | 0.75 |
| Freshwater Correction Value | Adjusted to conserve ocean seawater mass | 1.07945 | 1.07945 |

Tuning was performed by always continuing the previous tuning run. After updating the tuning parameters, new runs were continued from the end of the previous one to test new parameter choices, while assuming that the changes to the model configuration would only have an incremental effect. Only the first simulation started from climatological values based on Levitus as previously mentioned. This methodology limited the drift in each of the tuning experiments and reduced the need for computational resources. The concatenation of the retained tuning experiments in each iteration can be regarded as a long pre-spin-up for the coupled EC-Earth3-HR simulation.

After the long concatenated pre-spin-up run, we performed a 100-year spin-up with the final set of parameters, from which we started a 350-year pre-industrial control run. The time series of annual mean global temperature and AMOC, and September sea ice for the pre-industrial run are shown in Figure A1. Starting from the pre-industrial (PI) control run, we initiated a historical simulation spanning from 1850 to 2014. This simulation was extended with a future projection that continued until the year 2100, following the SSP2-4.5 emission scenario. The SSP2-4.5 scenario was selected because it is currently the CMIP6-Tier 1 scenario most closely aligned with $CO_2$ concentrations that could be reached following 'The Stated Policies Scenario' (STEPS), which provides an outlook based on the latest policy settings, including energy, climate and related industrial policies (IEA, 2023). In addition to the historical and future simulations, we also conducted a simulation with a 1% annual $CO_2$ increase starting from the PI control run. Despite the inclusion of this scenario, our study primarily focuses on the outcomes from the historical and future simulations. The results from these simulations contribute to understanding the potential impacts of climate change under moderate mitigation strategies and provide valuable insights into the trajectory of global climate dynamics over the next century.




## 3.    General evaluation of the model

As this is the first article to use the official EC-Earth3-HR version, we begin by providing an overview of the
model's performance and biases, focusing on key variables related to the atmosphere, ocean, and sea ice. This
overview sets the stage for a more detailed analysis of the North Atlantic and Arctic regions.

### 3.1 Mean historical climate

Figure 1 presents a comparison of the temperature time series of the EC-Earth3-HR model with reanalyses and
observational estimates. Furthermore, the results from the 24-member ensemble of the EC-Earth3-SR model for
CMIP6 are presented as a grey for reference. The global mean near-surface temperature (TAS) of the
high-resolution model is in good agreement with that of ERA5 (Hersbach et al., 2020), while the EC-Earth3-SR
ensemble consistently displays a higher temperature. The high-resolution model better depicts the observed
warming since 1950, while the  EC-Earth3-SR displays a trend stronger than that of ERA5. The situation
changes when the data are separated into land and ocean categories. Both model resolutions, exhibit temperatures
lower than that observed in both the CRU (Harris et al., 2020) and ERA5 datasets over land. EC-Earth3-HR
remains within the EC-Earth3-SR ensemble, but it is on the cold side. When compared to the CRU or ERA5
data, the EC-Earth3-HR model produces a temperature approximately 2°K lower than the observed values. In
contrast to land temperature, the global mean SST from the EC-Earth3-HR model is close to the value of the
HadISST data set (Rayner et al., 2003) while the SST in the EC-Earth3-SR ensemble is significantly warmer.

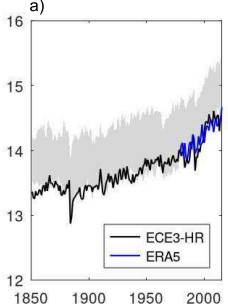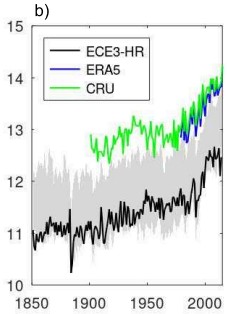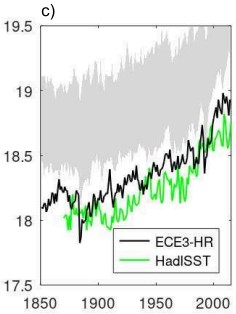


**Figure 1. Time series of near-surface air temperature (TAS, °C) and sea surface temperature (SST, °C) from EC-Earth3-HR (black), ERA5 reanalysis (blue), CRU over land (green), and HadISST over ocean (green) for the historical period. The full range of the EC-Earth3-SR CMIP6 historical ensemble is shown as a grey shaded area. Panels show: a) global mean, b) global mean over land (excluding Antarctica), and c) global mean over ocean.**


The present-day annual mean near-surface temperature (TAS), total precipitations (pr) and sea level pressure
(psl) are compared with ERA5 over the period 1980-2010 in Figure 2. All the parameters are regridded to 0.5° ×
0.5° using bicubic interpolation before comparison. Compared to ERA5, EC-Earth3-HR is colder over the Arctic
and warmer over Antarctica and the surrounding ocean areas. The extent to which EC-Earth3-HR exhibits a cold
bias in the Arctic remains unclear, given that ERA5 itself has a warm bias in the region (Tian et al. 2024).
EC-Earth3-HR is slightly colder over most of the continents, with the cold biases having a similar distribution
pattern to EC-Earth3-SR (Döscher et al., 2022). However, there are also noticeable warm biases over the
Siberian and the Greenland adjacent ocean regions. The Southern upwelling regions, along the South American



and African coasts, are better simulated in EC-Earth3-HR compared to EC-Earth3-SR with less warm bias. In
addition, the warm bias over Antarctica and the Southern Ocean has been previously attributed to the
parametrization of shortwave cloud radiative effects in climate models (Hyder et al., 2018; Dösher et al., 2022).
Similar to EC-Earth3-SR, the spatial precipitation pattern is well captured in EC-Earth3-HR. However, the
simulated double Intertropical Convergence Zone (ITCZ) is still displaced to the south, likely due to the
persistent strong warm bias over the Southern Ocean (Hwang and Frierson, 2013; Döscher et al., 2022). The wet
and dry biases in EC-Earth3-SR have been slightly improved in EC-Earth3-HR, particularly over the tropical
Pacific Ocean. For PSL, EC-Earth3-HR shows similar spatial patterns as ERA5. PSL biases are relatively small
in tropical and mid-latitude areas but are more pronounced over the Arctic and Antarctic. The substantial bias in
the Southern Ocean is primarily due to anomalous warming in our model simulation. In general, EC-Earth3-HR
reproduces a reasonable mean climate. TAS generally exhibits a bias of less than 1–2 K over most continents.
Regarding precipitation, the bias remains below 0.5 mm/day over most land areas, except in the Amazon region.
The improvements may result from a refined representation of orography.

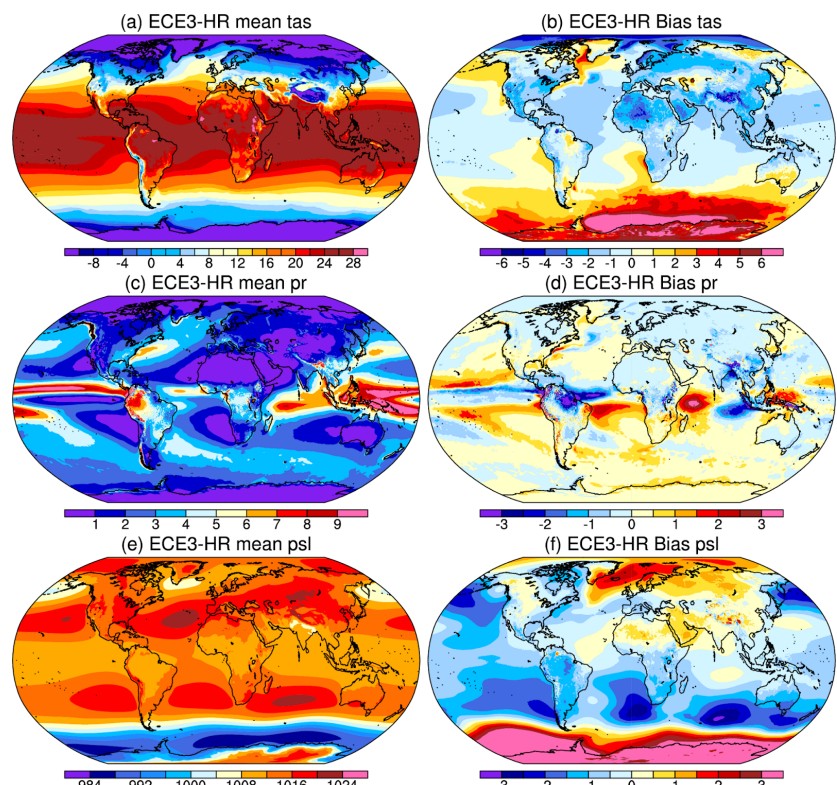

**Figure 2. Annual mean (a) near-surface temperature (TAS, °C), (c) total precipitations (pr, mm/day) and (e) sea level**
**pressure (psl, hPa) simulated by EC-Earth3-HR for the period 1980-2010, and (b), (d), (f) their respective biases**
**compared to ERA5.**

There are large regional differences in sea ice concentration between EC-Earth3-HR and satellite observations,
and these differences are generally similar to those observed in EC-Earth3-SR (Figure 3a,b). In March, the model




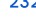

**Figure 3. (a, b) Difference in sea ice concentration (%) between the EC-Earth3-HR model and satellite observations, averaged over 1980–2010; (a) shows September, (b) shows March. (c, d) Sea surface temperature (SST) bias (°C) compared to HadISST in EC-Earth3-HR and the ensemble mean of EC-Earth3-SR, respectively. (e, f) Sea surface salinity (SSS) bias (psu) compared to the WOA18 climatology in EC-Earth3-HR and the ensemble mean of EC-Earth3-SR, respectively.**


overestimates sea ice concentration in the Greenland-Iceland-Norway (GIN) and Barents Seas, and underestimates it in the Labrador and Bering Seas. The largest discrepancy between EC-Earth3-HR and



EC-Earth3-SR (see Figure 11 in Döscher et al., 2022) occurs in the sea ice coverage over the Labrador Sea likely
due to the difference in fluxes through CAA (Karami et al. 2021). In September, the sea ice concentration is
underestimated in the Kara, Laptev and Chukchi Seas, while exhibiting an overestimation at the ice margins.
Different regions of the Arctic are losing sea ice at different rates. The trend in sea ice concentration (% per
decade) from 1980-2010 compared to observations, shows notable regional discrepancies, particularly in the
Beaufort and Chukchi seas (Figure A2).

Regarding SST bias, Figure 3c,d  illustrates the bias between EC-Earth3-HR and HadISST, and compares it to
the bias between the ensemble mean of EC-Earth3-SR and HadISST. The patterns are generally similar in both
resolutions. The largest biases occur along the east coast of North America and in the southern part of the
subpolar gyre and reach up to around ±5 °C. These biases are typical of standard resolution climate models
where the Gulf Stream extends too far to the north along the North American coast and the North Atlantic
Current goes too zonally across the North Atlantic, thus allowing the subpolar gyre to extend too far to the south.
EC-Earth3-HR shows a somewhat less bias in the Gulf Stream region along the North American east coast. This
is linked to improvements in the position of the Gulfstream and the North Atlantic Current in the high-resolution
version. On the other hand, EC-Earth3-HR shows a positive bias in the Labrador Sea, mainly due less sea ice in
this region. Also in the subtropical North Atlantic, a slightly larger cold bias is visible in EC-Earth3-HR
compared to EC-Earth3-SR. It should be noted that the biases observed in individual EC-Earth3-SR simulations
are generally more pronounced than those evident in the ensemble mean that is used here as comparison. The
bias pattern of SSS (Figure 3e,f) in EC-Earth3-HR is comparable to that of the SST bias. In most areas with a
warm bias (Labrador Sea, along North american coast), EC-Earth3-HR simulates too high salinity compared to
the observational based climatology while in the cold-bias regions, salinity tends to be too low. As previously
seen for the SST-biases, the SSS-biases in EC-Earth3-HR are smaller in the Gulf-Stream - North Atlantic Current
region than in EC-Earth3-SR. In contrast to EC-Earth3-HR, EC-Earth3-SR also simulates too low salinity in the
entire Labrador Sea and Baffin-Bay area. Overall, compared to EC-Earth3-SR, the improvements strongly
depend on the geographical region. The enhanced horizontal resolution does not consistently yield improvements
across all processes due to inherent limitations in parameterization (Fosser et al., 2015).

**3.2 Climate variability modes**
Figure 4 presents a comparison of the NAO patterns in EC-Earth3-HR with those in EC-Earth-SR and in ERA5.
Both EC-Earth3-versions exhibit similarities in the pattern and location of the high and low centers, closely
aligning with the reference ERA5 plot. However, EC-Earth3-HR underestimates the intensity of both the positive
and negative pressure centers, which are located in the subtropical and North Atlantic regions, respectively.
We calculated the regression of the Niño 3.4 index (5°N-5°S, 170°W-120°W) on sea level pressure for the winter
season (December to February, DJF) using ERA5 reanalysis data and the EC-Earth3-SR and EC-Earth3-HR
models (Figure 5a,d,g). The results reveal a positive Pacific North American (PNA) like pattern characterized by
an intensified low-pressure system over the Aleutian region and lower pressure over the eastern United States.
This low-pressure system extends along the North American east coast into the central North Atlantic Ocean,
while higher pressure is observed over and between Iceland and Scandinavia, resembling a negative phase of the
NAO. Both EC-Earth3-SR and EC-Earth3-HR show strong similarity to ERA5, with the high-resolution model
exhibiting higher intensity and greater resemblance to ERA5. Moreover, EC-Earth3-SR shows weaker surface



pressure anomalies, likely due to a slightly weaker ENSO intensity compared to EC-Earth3-HR. Therefore,
EC-Earth3-HR represents a clear improvement in capturing the ENSO-SLP response. The SST Niño 3.4 index
anomaly spectrum exhibits dominant variability in the 4-6 year range (Figure 5c,f,i). EC-Earth3-SR captures the
most significant frequencies similarly to HadISST, while EC-Earth3-HR displays higher intensity at the 4-year
frequency.

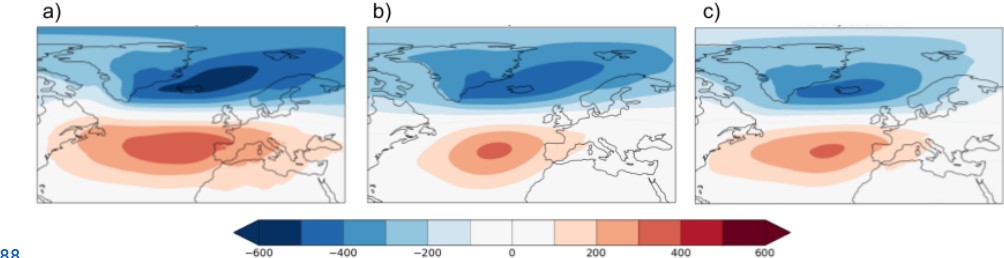


**Figure 4. Regression of sea level pressure anomalies onto the NAO index during winter (December–February) for ERA5 reanalysis (a), EC-Earth3-SR member 1 (b), and EC-Earth3-HR (c). The NAO index is defined as the difference in area-averaged SLP between the Azores (31.5°W–24.5°W, 36.5°N–40°N) and Iceland (25°W–13°W, 63°N–67°N), following Fereday et al. (2020) and Fuentes-Franco et al. (2023).**
293

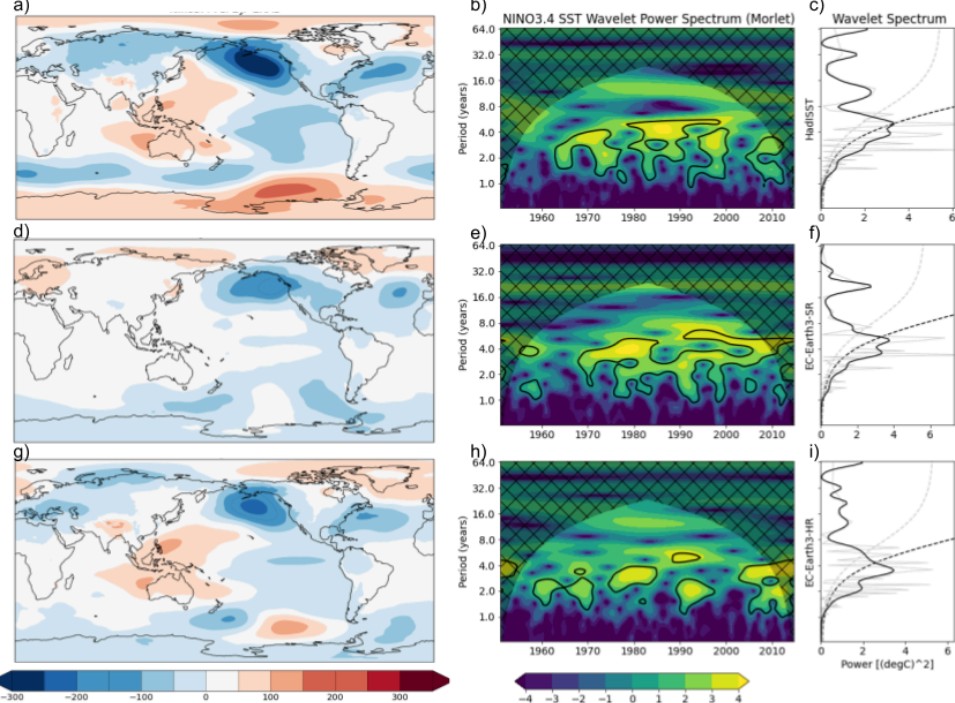

294

**Figure 5. Niño 3.4 analyses during winter (DJF): regression of sea level pressure anomalies onto the Niño 3.4 SST index (left column), wavelet power spectra (middle column), and global wavelet power spectra (right column). Rows correspond to different datasets: ERA5 (top row: a–c), EC-Earth3-SR member 1 (middle row: d–f), and EC-Earth3-HR (bottom row: g–i). The Niño 3.4 index is defined as the average SST over 5°N–5°S, 170°W–120°W. Power spectra are computed over 1951–2014 using 3-month means for December–February.**
300



**4. Projected changes in the Arctic and North Atlantic Oceans under the SSP2-4.5 scenario**

**4.1 Changes in the Arctic sea ice and temperature in future projections**

Figure 6a shows the time series of September Arctic sea ice area from 1850 to 2100 for EC-Earth3-HR. It slightly underestimates the Arctic sea ice area compared to the sea ice reconstruction of Walsh et al. (2019) and the OSI SAF satellite observations (Lavergne et al., 2019). However, it captures the rapid observed decline of sea ice between 1995 and 2010. While these results are based on a single-member simulation, making it difficult to draw firm conclusions, it still highlights the ability of the model to simulate realistic, rapid sea ice loss. The sea ice area shows a significant downward trend, with a marked reduction starting around 2000. From 2040, the Arctic becomes nearly ice-free in September, with sea ice area consistently below 1 million km², reaching fully ice-free conditions by 2050 under the SSP2-4.5 scenario. The projected timing of an ice-free Arctic varies widely across models, reflecting large uncertainty. In CMIP6, the multimodel ensemble mean projects September ice-free conditions by 2050 under the high-emission scenario SSP5-8.5, with delayed occurrences for SSP2-4.5 (Notz and SIMIP Community, 2020). However, both Docquier and Koenigk (2021) and Selivanova et al. (2024) project an ice-free Arctic around before 2040 based on observationally constrained model projections, consistent with our EC-Earth3-HR estimate.

The decline in sea ice is closely linked to rising Arctic temperature, which is increasing faster than the global average—a phenomenon known as Arctic amplification (Holland and Landrum, 2021). We find that the annual mean global surface temperature anomaly and the Arctic-averaged temperature anomaly (both relative to modeled pre-industrial values) exhibit distinct warming trends, as expected. In EC-Earth3-HR, the global mean temperature anomaly (relative to PrI) reaches 1.5°C by around 2023, 2°C by 2045, and 3.2°C by 2100, while the Arctic experiences much stronger warming, with an anomaly of 2°C by around 2000, accelerating to 4°C by 2040 and 7°C by 2100 (Figure 6b). The EC-Earth3-SR ensemble confirms this pattern but shows stronger global and Arctic warming than EC-Earth3-HR throughout the 21st century.

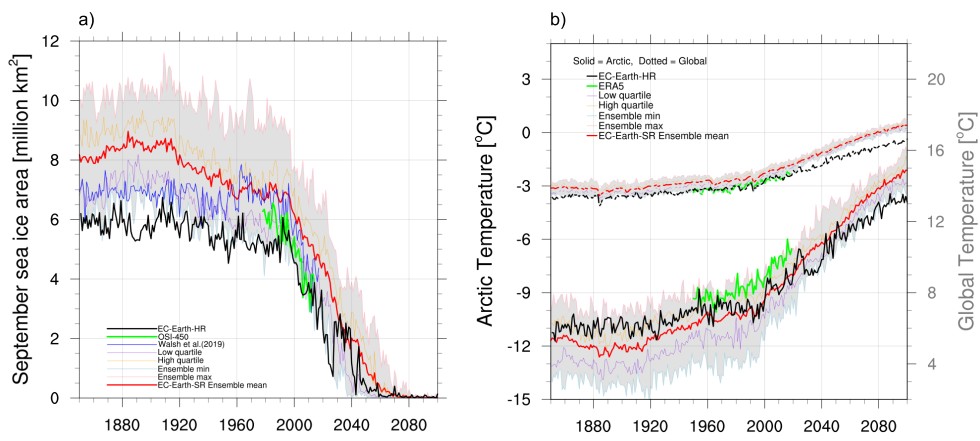

**Figure 6. (a) September Arctic sea ice area (million km²) from 1850 to 2100 based on EC-Earth3-HR (black), the 14-member EC-Earth3-SR ensemble (red: mean; grey shading indicating the ensemble spread including minimum, maximum, and interquartile range), and satellite/reconstructed observations: OSI-450 (green) (OSI SAF, 2017) and sea ice reconstruction of Walsh et al. (2019) (blue). (b) Annual mean near-surface air temperature (°C) over the Arctic (solid lines) and the globe (dotted lines) for EC-Earth3-HR (black), ERA5 reanalysis (green), and the EC-Earth3-SR ensemble (red: mean; grey shading as in panel a). Historical simulations cover 1850–2014, with projections from 2015–2100 under the SSP2-4.5 scenario.**



To illustrate the projected spatial sea ice changes, we compare the March sea ice concentration for the 2070-2100
mean with the 1980-2010 mean (Figure A3). Substantial declines are observed, particularly in the Barents,
Greenland and Bering Seas. As September sea ice is projected to disappear by mid-century, we focus instead on
the trend from 2020 to 2050 for this month. The trend in September sea ice concentration (% per decade) shows
that sea ice loss is occurring at different rates in different Arctic regions with particularly strong reductions in the
Central Arctic Basin and the Canadian Arctic Archipelago. These spatial variations indicate that sea ice loss will
be more pronounced in certain areas in the coming decades.

**4.2 Projected changes in North Atlantic Ocean properties**

The spatial distribution of ocean temperature and salinity, through its influence on the zonal and meridional
density gradients, is an important driver of ocean circulation changes. Furthermore, the distribution of North
Atlantic sea surface temperatures can impact the large scale atmospheric circulation (e.g. Gastineau &
Frankignoul, 2015). The projected change in SST, comparing the 2070-2100 mean to the 1980-2010 mean,
shows moderate to strong warming (Figure 7a), though this warming is not uniform across all regions. The most
pronounced warming occurs where the winter sea ice edge has shrunk, particularly in the Greenland Sea. In
comparison, EC-Earth3-SR (Figure A4) exhibits more pronounced warming than EC-Earth3-HR, with a broader
extent across the entire Nordic Seas. EC-Earth3-SR also shows considerably greater warming in the Labrador
Sea and the subpolar gyre. This difference between the two model versions is partly due to the larger extent of
sea ice in EC-Earth3-SR, which led to more significant sea ice reductions in these regions. The North Atlantic
warming hole, a region characterized by no temperature increase in the eastern subpolar gyre (e.g., Liu et al.,
2020), is also evident in Figure 7a, where a moderate surface cooling anomaly can be seen compared to the
historical period. This cooling pattern has been linked to the weakening of the AMOC (Drijfhout et al., 2012).
Looking at the region of the warming hole in the deeper layers, we find that the cooling is more present in the
upper few hundred meters (not shown).

The projected changes in SSS in EC-Earth3-HR (Figure 7b) are dominated by decreasing salinity in mid and
high latitudes, with exceptions in the Greenland Sea and eastern Arctic, where salinity increases. The freshening
is most pronounced in the Beaufort Gyre and western Arctic. Conversely, salinification in the Greenland Sea and
eastern Arctic is also projected. Salinity changes may arise from changes in Arctic circulation (Karami et al.,
2023), river runoff and precipitation, and increased advection of Atlantic Water, but the underlying mechanisms
remain uncertain and require further investigation beyond the scope of this study. EC-Earth3-SR exhibits an
overall similar pattern of SSS change (Figure A4). This model agreement supports the robustness of the
projected SSS changes across resolutions, despite model uncertainties.

The distribution of water masses in the North Atlantic strongly influences mixing and deep-water formation,
particularly in the core convection regions of the Labrador, Irminger, and Greenland Sea. To better understand
these changes, we analyze sea surface density (SSD) and mixed layer depth (MLD) for the winter season. The
SSD pattern resembles salinity over the Arctic, with positive anomaly in the eastern Arctic and negative anomaly
in the western Arctic (Figure 7c). This density contrast reflects the dominance of salinity-driven changes in the
surface Arctic Ocean. Over the North Atlantic, both SST and SSS contribute to density changes, though SST



plays a dominant role in driving SSD distribution over the central to eastern Atlantic (east of 30°W). The
influence of SST is particularly evident in areas with strong surface warming, which reduces density and vertical
mixing. The winter mixed layer depth serves as a proxy for the location and depth of deep convection, and the
MLD analysis reveals a clear reduction of its intensity in the Labrador, Irminger, and Greenland Seas (Figure
7d). This is spatially consistent with SSD reductions in these regions that has led to increased stratification and
weakened convective mixing. These patterns are particularly evident in the Labrador and Greenland Seas,
indicating that there may be regionally distinct drivers of stratification. In the former region, there is a
combination of warming and freshwater accumulation; in the latter region, warming is the predominant factor.
This decrease in MLD, driven by density changes in the North Atlantic, is linked to reduced deep-water
formation and contributes to a slowdown of the AMOC. These changes in the AMOC, including its present-day
structure and projected future weakening, are discussed in detail below.

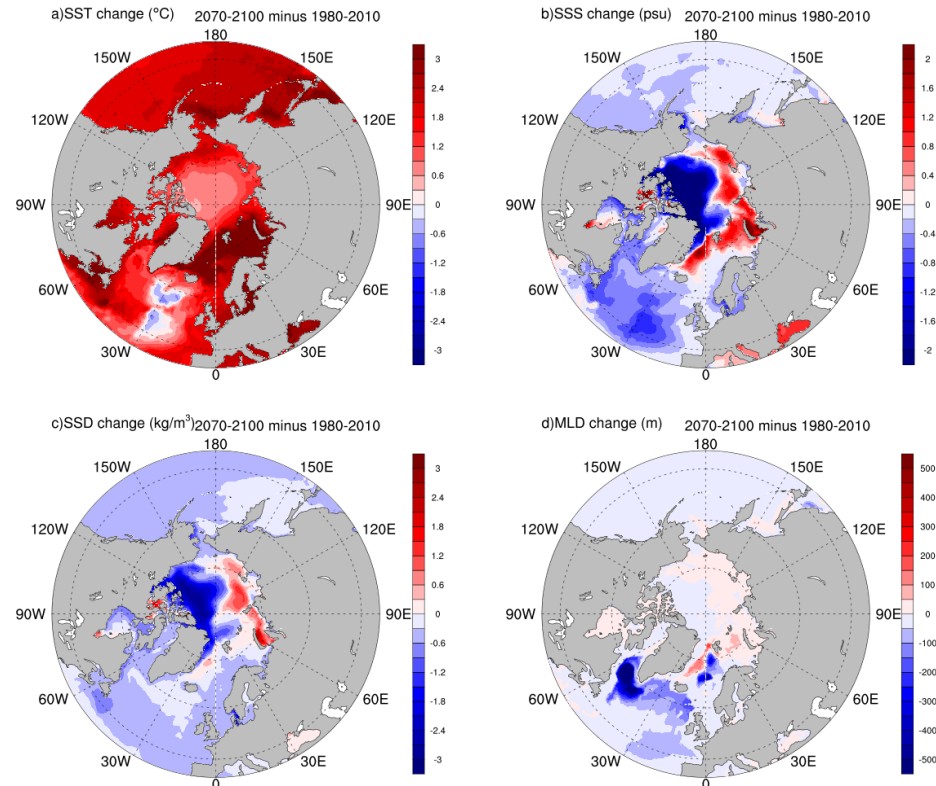


**Figure 7. Differences between the 2071–2100 and 1980–2010 periods under the SSP2-4.5 scenario in EC-Earth3-HR**
**for: (a) annual mean sea surface temperature (SST, °C); (b) annual mean sea surface salinity (SSS, psu); (c) winter**
**mean (JFM) sea surface density (SSD, kg/m³); (d) winter mean (JFM) mixed layer depth (MLD, m). All figures are**
**shown for 40°N northward.**

## 4.3 Changes in the AMOC

AMOC involves the northward flow of warm, salty water in the upper Atlantic Ocean, which cools and sinks at
high latitudes, driving a southward flow of cold, dense water in the deeper layers (Buckley and Marshall, 2016).



The AMOC stream function reveals the typical overturning circulation (Figure 8) and, for the 1980–2010 mean,
has a peak transport of more than 18 Sv occurring at a depth of approximately 1000 m. Compared to the
ensemble mean of EC-Earth3-SR used in CMIP6 (Figure 15 in Döscher et al., 2022), EC-Earth3-HR simulates a
slightly stronger AMOC. The AMOC stream function is projected to weaken in the future, with the mean for the
period 2070–2100 showing a decline of 5.6 Sv relative to the 1980–2010 mean (Figure 8c). This AMOC
weakening is consistent with the reductions in density and MLD discussed earlier. The projected structure of the
overturning cell also exhibits a shallower and less extensive circulation, reflecting disruptions to key driving
mechanisms, such as dense water formation. This reduction of AMOC under future climate scenarios aligns with
CMIP6 model projections (Bellomo et al., 2021).

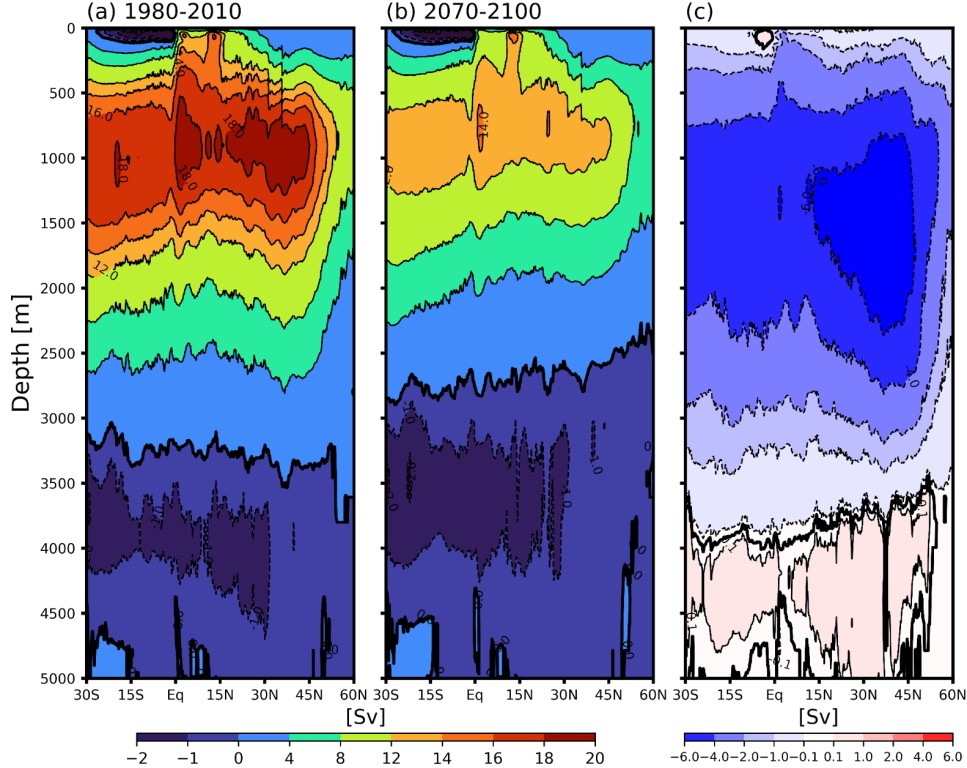


**Figure 8. AMOC stream function (Sv) in the depth-latitude plane for EC-Earth3-HR, averaged over 1980–2010 (a)
and 2070-2100 (b) and their difference (c).**

The time series of the AMOC index, calculated as the maximum volume transport stream function between 20°N
and 40°N and 800–1100 m depth (Figure 9a), is slightly higher than the observations from the RAPID-MOCHA
array (Smeed et al., 2018). However, the model captures variability of comparable amplitude to observations.
The EC-Earth3-HR simulation closely follows the ensemble maximum values of EC-Earth3-SR until around
year 2000 but remains close to the ensemble mean after that. The AMOC index shows a clear reduction in the
21st century, decreasing from around 21 Sv to 13 Sv, which represents a reduction of approximately 39%. This is
within the range of the findings of Weijer et al. (2020), where they estimated a potential decline of 6 to 8 Sv
(34–45%) of AMOC in CMIP6 models, after selecting the models constrained with RAPID observations.



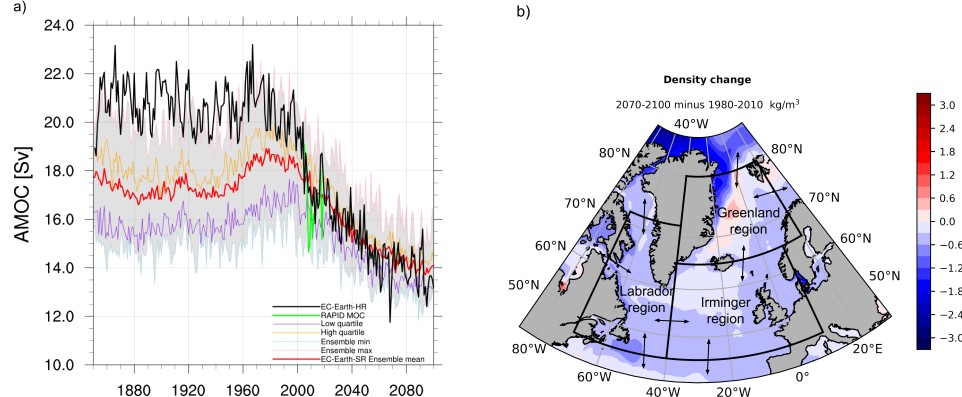

**Figure 9. (a) Annual-mean AMOC strength (maximum overturning streamfunction between 20°–40°N at 800–1100 m depth) from 1850 to 2100 based on EC-Earth3-HR (black), the 14-member EC-Earth3-SR ensemble (red: mean; grey shading indicating ensemble spread including minimum, maximum, and interquartile range), and RAPID-MOCHA observations (green) (Smeed et al., 2018). (b) Winter mean density change (kg/m³/decade) between the 2071–2100 and 1980–2010 periods in the North Atlantic showing the three analysis regions used in Section 4.4: the Greenland, Irminger, and Labrador regions. These boxes represent extended areas encompassing and surrounding the respective seas. Arrows indicate the sections where inflows and outflows are calculated.**

### 4.4 Processes driving the weakening of the AMOC

The North Atlantic plays a critical role in AMOC dynamics through deep convection and the formation of dense water masses. These processes primarily occur in three key regions: the Labrador Sea, the Irminger Sea, and the Greenland Sea (Figure 9b). To assess changes in these regions in EC-Earth3-HR, we first examine the time–depth evolution of density (Figure 10), which highlights long-term changes in stratification and isopycnal structure. This analysis is complemented by two indices: one representing convection and the other quantifying the rate of deep water formation.

- **Increased stratification**

Figure 10 shows the evolution of density (kg m⁻³) profiles averaged over the Greenland, Irminger, and Labrador regions (shown in Figure 9b) from 1851 to 2100, highlighting long-term changes in vertical stratification in the North Atlantic. In all three regions, stratification becomes increasingly pronounced over time, particularly in Greenland and Labrador after the late 20th century, as lighter density classes expand and thicken in the upper ocean. These trends reflect enhanced surface stratification, which inhibits vertical mixing and suppresses deep convection, consistent with changes in surface forcing and water mass transformation under climate change. In the Greenland Sea, the upper ocean shows a marked increase in the thickness of low-density layers, especially after the year 2000, while the denser water masses at depth retreat and become less prominent. In the Irminger region, lighter densities gradually occupy a greater proportion of the upper 1000 m over time. In the Labrador region, a similar pattern emerges, with a pronounced thickening of lighter density classes in the upper ocean and a corresponding reduction in the volume of dense water at depth.

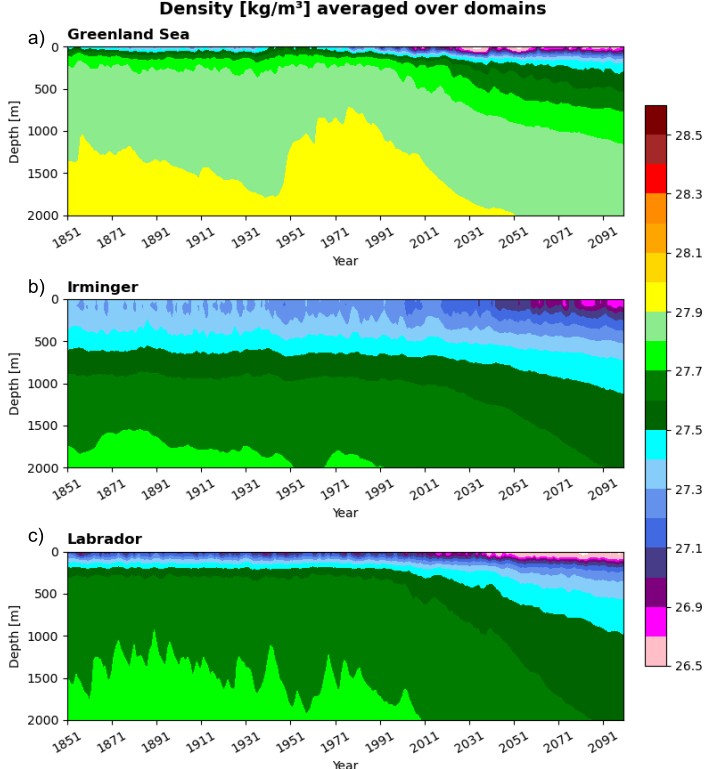

443

**Figure 10. Time–depth evolution of density (kg m⁻³) averaged over the Greenland, Irminger, and Labrador regions (shown in 9b) from 1850 to 2100. Density is using winter (DJF) values from historical and scenario simulations, with color shading indicating isopycnal structure. The regions correspond to extended boxes shown in Figure 9b.**

- **Weakened convection**

A common index to diagnosing convection is the MLD, but it only provides depth information and does not capture the spatial extent of convection. Here, we use the Deep Mixed Volume (DMV) index (Brodeau and Koenigk, 2016), which quantifies the total volume of water involved in deep convection by integrating both the depth and horizontal extent of the mixed layer thus offers a more comprehensive measure of convection. The DMV integrates the volume of mixed water masses below a critical depth over the Labrador, Irminger and the Greenland regions (shown in Figure 9b), thereby representing the extent of deep convection in these key regions. Here, we chose the same depth criterion of 1000 m for the Labrador and Irminger Seas and 700 m for the Greenland Sea as done in Brodeau and Koenigk (2016).

The DMV in EC-Earth3-HR shows notable internal variability across all three basins until the late 20th century, followed by a marked and persistent decline (Figure 11). The Labrador Sea exhibits the most pronounced collapse, with DMV approaching near-zero by 2050, while the Greenland Sea begins its decline earlier, and shuts down around 2020. The Irminger Sea displays consistently weak convection, with low DMV values and only occasional minor peaks throughout the time series, and undergoes a sustained reduction. This widespread DMV reduction reflects a substantial weakening of deep convection, consistent with the projected AMOC slowdown.



The correlation coefficient between the Labrador Sea DMV and AMOC is 0.84, the Greenland Sea DMV and
AMOC is 0.67, and the Irminger Sea DMV and AMOC is 0.35 over the entire time series. When the DMV
values are combined, the correlation with AMOC increases to 0.87. This suggests that the future AMOC
reduction is more closely linked to the combined DMV trend in Labrador, Irminger and Greenland Seas
convection. The maximum correlation was found at lag zero, and no significant lag was observed in these
correlations.
Sensitivity analysis using different critical depth thresholds reveals that with a shallower criterion (500 m), the
Irminger Sea maintains moderate convective area into the mid-21st century. This suggests that convection in the
Irminger Sea becomes progressively shallower rather than shutting down entirely. In contrast, both shallow and
deep MLD thresholds in the Labrador and Greenland Seas decline nearly synchronously, indicating an abrupt and
near-total collapse of convection in these regions (Figure A5). When compared to EC-Earth3-SR, the DMV in
EC-Earth3-HR is much stronger, consistent with findings from Koenigk et al. (2021), which indicate an
enhanced DMV with increasing resolution in HighResMIP models. However, despite the higher DMV values in
EC-Earth3-HR, its decline starts slightly earlier than in EC-Earth3-SR. In EC-Earth3-HR, convection in the
Greenland Sea shuts down first, followed by the Labrador Sea after a delay of about two decades. In contrast, in
EC-Earth3-SR, the DMV shuts down earlier in the Labrador Sea and later in the Greenland Sea.

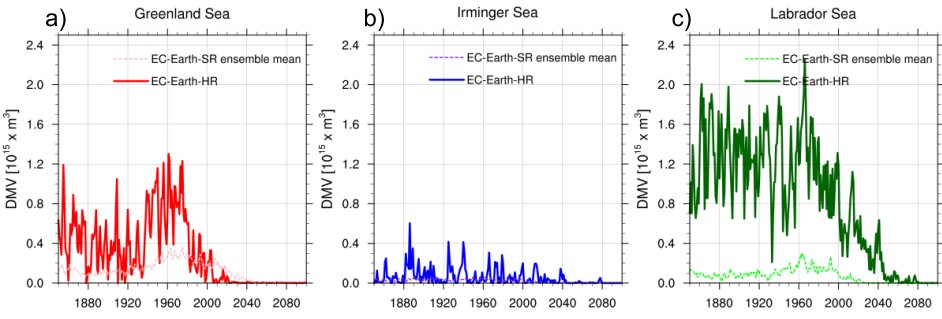


**Figure 11. Deep Mixed Volume (DMV) in the Greenland (a), Irminger (b), and Labrador (c) Seas from historical and**
**SSP2-4.5 simulations of EC-Earth3-HR (thick lines) and the ensemble mean of EC-Earth3-SR simulations (dashed**
**lines).**

**- Reduced deep water formation**
Although the DMV index reflects the evolution and potential disappearance of deep convection at key locations
in our model, it does not directly quantify the rate of deep water formation and relative contribution of each of
the deep water formation sites. To complement the DMV, and in line with the reduced stratification and MLD
discussed above, we examine changes in deep water formation (DWF) and their contribution to AMOC
weakening. To this end, we propose a volume-conservation-based method to estimate DWF within the same
three key regions as before (Figure 9b). This approach provides an approximation and serves as an index of
DWF. Each region is divided into two layers: an upper layer from the surface to the critical depth (as defined
above), and a lower layer from the critical depth to the seafloor. For each layer, we calculate the net volume
transport of water entering and exiting through the boundaries of the upper and lower layers (Figure 12a). In all
three regions, the analysis reveals a net inflow of water into the upper layer and a corresponding net outflow





from the lower layer. This pattern indicates convergence within the upper layer, driving sinking and downward
transfer of water to the lower layer. We interpret this vertical water movement as a representation of the DWF,
which ultimately feeds into and sustains the AMOC.

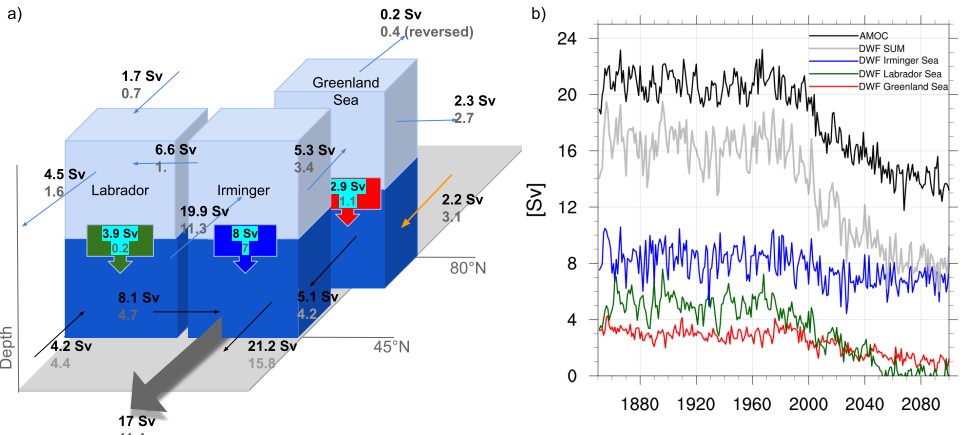


**Figure 12. (a) Illustration of the regions used for DWF calculations: blue arrows represent inflows and outflows in the upper layers, and black arrows indicate flows in the lower layers. Colored callouts with the downward arrows display DWF, with values in black corresponding to the 1980–2010 mean and numbers in grey to the 2070–2100 mean. The large grey arrow represents the net flow through 45°N, calculated as the sum of all DWFs and the deep inflow from the Arctic into the Greenland Sea. The orange arrow shows the deep outflow from the Arctic into the Greenland Sea. (b) Time series of the AMOC (black) and deep water formation (DWF) rates in the Irminger region (blue), Labrador region (green), and Greenland region (red), along with the total DWF (DWF SUM, grey) from 1850 to 2100, based on EC-Earth3-HR. All values are in Sverdrups [Sv]. A long-term decline is evident in AMOC and DWF, particularly after the year 2000.**


The inflow into the Irminger region follows three pathways: (1) partially sinking as DWF within the Irminger
region, (2) flowing into the Labrador region where it contributes to the DWF before returning to the Atlantic, and
(3) flowing into the Greenland Sea where it contributes to DWF (Figure 12). These DWFs feed into the AMOC.
The sum of the three DWF rates plus the Arctic deep flow into the Greenland Sea equals the net deep southward
flow at 45°N (represented by the grey thick arrow in Figure 12a), which averages about 17 Sv over 1980–2010.
The Irminger Sea box has the largest DWF contribution to this net flow in our model—approximately
47%—followed by the Labrador Sea (23%) and the Greenland Sea (17%), a pattern that remains consistent over
the historical period. This dominant role of the Irminger Sea is consistent with Lozier et al. (2019), and our
estimate for Labrador Sea closely aligns with that of Zou et al. (2020) who calculated transports across density
layers. Notably, the Labrador Sea stands out as the primary contributor to AMOC weakening, experiencing the
largest DWF decrease in future projection. Comparing the 2070–2100 mean to 1980–2010, DWF reduces from
3.8 Sv to 0.1 Sv in Labrador, approaching collapse, while it decreases from 8 to 6.9 in the Irminger and from 2.8
to 1 in the Greenland Sea (Figure 12). The total DWF (the sum of all three regions) decreases by −6.5 Sv, from
14.8 Sv in 1980–2010 to 8.3 Sv in 2070–2100, slightly exceeding the AMOC reduction of −5.6 Sv over the same
period. Interestingly, this difference is compensated by an increased deep inflow of +0.9 Sv from the Arctic into
the Greenland Sea, as shown in our analysis. These results suggest that DWFs in the Labrador and Greenland



Seas are the primary driver of long-term AMOC weakening under climate change, whereas the Irminger DWF,
with the strongest contribution to the AMOC, plays a crucial role in sustaining it.
In addition to these mean changes, the temporal covariability between DWF and the AMOC index provides
further insight into the underlying dynamics. Over the entire simulation period (1850–2100), the AMOC index is
significantly correlated with DWF in all three regions: r = 0.93 for the Labrador Sea, r = 0.64 for the Irminger
Sea, and r = 0.82 for the Greenland Sea. Notably, the Labrador Sea shows the strongest correlation, further
supporting its role as the primary driver of AMOC weakening. However, this relationship is largely shaped by
anthropogenic warming trends, as the correlations are weaker over the sub-period 1850–1980. During that
sub-period, the correlations decline across all regions, with r = 0.42 for the Labrador Sea, r = 0.53 for the
Irminger Sea, and r = –0.10 for the Greenland Sea, indicating weaker coupling. When the total DWF is
considered, its correlation with the AMOC index exceeds that of any individual region: r = 0.94 over 1850–2100
and r = 0.65 over 1850–1980. This strong relationship highlights the critical role of cumulative DWF in
modulating AMOC strength.

**5. Discussion and Conclusion**
This study focuses on the Arctic–North Atlantic climate using the high-resolution EC-Earth3-HR model,
representing both historical and future climate states. Haarsma et al. (2020) reported that the previous version of
the model, EC-Earth3P-HR, did not reduce most model biases compared to its lower-resolution counterpart,
EC-Earth3-SR, and, in some cases, even degraded performance, attributing this to the absence of re-tuning and
the short spin-up period mandated by the HighResMIP protocol. Here, we address these issues through extensive
tuning and longer spin-up of EC-Earth3-HR, resulting in improved simulation performance. Compared to
EC-Earth3P-HR simulations under HighResMIP, which suffered persistent biases and oceanic drift (Haarsma et
al., 2020; Roberts et al., 2019), the updated configuration shows reduced radiative imbalances and minimal
deep-ocean drift. These improvements lower uncertainties in long-term trends and internal variability, enabling
more robust attribution of future changes to model resolution rather than to artefacts of initialisation or
configuration. Our simulations also include full historical and scenario integrations through 2100, extending
beyond the constraints of the HighResMIP protocol.
While EC-Earth3-HR shows only modest improvements in climatological means of key variables (e.g., SST,
surface air temperature, precipitation, and sea ice) compared to EC-Earth3-SR, it exhibits notable improvements
in simulating climate variability and trend. Time series of global and Arctic temperatures, AMOC strength, SST
and sea ice area show better agreement with observations and reanalysis, indicating that higher resolution
enhances the model's ability to capture transient responses and low-frequency variability. While higher
resolution is expected to improve atmospheric teleconnections, extreme events, ocean currents and fluxes
through key gateways, these aspects were not explicitly assessed here and could be better investigated via
ensemble simulations (Roberts et al., 2020). Overall, the improved representation of time-evolving climate
processes in EC-Earth3-HR highlights the value of high-resolution modeling, while a full evaluation of its
advantages calls for further process-based analysis.



The improved performance of the EC-Earth3-HR model in simulating Arctic–North Atlantic climate extends to
key components such as Arctic sea ice, deep convection, and AMOC variability. EC-Earth3-HR reproduces both
the amplitude of observed variability and the timing of rapid decline in Arctic sea ice and the AMOC, indicating
its capability to realistically simulate changes driven by the interplay of anthropogenic forcing and internal
variability. The model simulates a smaller ice area than EC-Earth3-SR, consistent with Docquier et al. (2019)
based on a subset of HighResMIP models, and exhibits a stronger sea ice decline, particularly toward the end of
the historical period and under SSP2-4.5. Summer sea ice area falls below 1 million km² around 2040, aligning
with observationally constrained projections (Docquier and Koenigk, 2021; Selivanova et al., 2024). Regarding
AMOC, the model reproduces the amplitude of observed AMOC variability and projects a stronger 21st-century
decline—about 33% in EC-Earth3-HR versus 25% in EC-Earth3-SR (Wyser et al., 2021)—confirming earlier
HighResMIP findings (Roberts et al., 2020, Jackson et al., 2020). This larger AMOC reduction leads to a more
pronounced North Atlantic warming hole and reduced warming along western European coasts, both of which
are evident in our results. These features may influence future changes in atmospheric circulation, blocking, and
extreme events over Europe—topics that will be addressed in future work.
EC-Earth3-HR also simulates strong deep convection in the Labrador and Greenland Seas during the historical
period, substantially stronger than in EC-Earth3-SR and consistent with findings from HighResMIP models
(Koenigk et al., 2021). This enhanced deep convection supports a stronger AMOC in EC-Earth3-HR during the
historical period relative to the standard-resolution version. This is while the horizontal and vertical distribution
of water masses improves in EC-Earth3-HR. Using the DMV index as a proxy, deep convection in
EC-Earth3-HR rapidly weakens from the late 19th century, ceasing entirely by ~2020 in the Greenland Sea and
~2050 in both the Labrador and Irminger Seas. This decline is linked to increased stratification due to surface
warming and freshening. However, while the DMV index effectively captures the presence and breakdown of
deep convection, it does not directly quantify deep water formation rates or the relative importance of individual
formation sites. Moreover, our modelling suggests that despite DMV's relevance for AMOC variability and
decline, approximately two-thirds of the AMOC strength persists even after a complete cessation of deep mixing
in the North Atlantic—indicating that DMV and MLD may not be sufficient proxies for AMOC strength.
To overcome this limitation and further investigate the drivers of AMOC weakening, we developed a novel
volume-conservation-based index for deep water formation (DWF). Although simplified, our method offers a
novel and accessible framework for estimating DWF, emphasizing the value of volume conservation principles
and providing process-level insights into variability in DWF—a key control on AMOC strength. Our DWF
index, offers a complementary perspective on deep water formation, particularly where proxies like DMV fall
short.In future projections, our results show that deep water formation in the Labrador Sea undergoes a complete
shutdown, while it declines by 62% in the Greenland Sea and only 13% in the Irminger Sea. Notably, while the
Labrador Sea dominates the weakening trend, the Irminger Basin plays a crucial role in sustaining the AMOC,
consistent with its dominant contribution to the overturning circulation. This approach could complement other
observational and modeling techniques for assessing deep water formation rates. Future refinement, including
adjustments to box definitions, critical depths, and validation against observational datasets, would enhance its
robustness and applicability.




Overall, the results underscore the potential of high-resolution modelling to capture key aspects of climate
variability and change in the Arctic–North Atlantic sector. Future work should focus on assessing extremes,
teleconnections, and feedbacks using ensemble of simulations and process-based studies to fully exploit the
potential of high-resolution simulations.

**Data availability**
Data from the EC-Earth3-HR historical and SSP2-4.5 simulations are available through any ESGF data node as
part of CMIP6; please search for 'EC-Earth3-HR'.
The reconstructed sea ice data of Walsh et al (2019) is downloaded from:
https://nsidc.org/data/g10010/versions/2.
The OSI-450 satellite data is downloaded from:
https://navigator.eumetsat.int/product/EO:EUM:DAT:MULT:OSI-450.
The CRU data is available via: https://crudata.uea.ac.uk/cru/data/hrg/.
HadISST data can be downloaded from: https://www.metoffice.gov.uk/hadobs/hadisst/
ERA5 data is available via: https://www.ecmwf.int/en/forecasts/dataset/ecmwf-reanalysis-v5

**Code availability**
EC-Earth: The EC-Earth model is restricted to institutes that have signed a memorandum of understanding or
letter of intent with the EC-Earth consortium and a software license agreement with the ECMWF. Confidential
access to the code and to the data used to produce the simulations described in this paper can be granted for
editors and reviewers; please use the contact form at http://www.ec-earth.org/about/contact

Calculation of Deep Water Formation (DWF) analysis:
CDFTOOLS was used to compute volume transport across various cross-sections. CDFTOOLS is a diagnostic
package developed within the DRAKKAR framework for analyzing NEMO model output:
https://github.com/meom-group/CDFTOOLS

**Acknowledgments**
This study was supported by the Horizon Europe project OptimESM "Optimal High Resolution Earth System
Models for Exploring Future Climate Changes" under the European Union's Horizon Europe research and
innovation programme (grant agreement No 101081193), the FORMAS project FutureGS (Grant 2021-01374),
the Swedish Research Council grant VR (2020-04791) and Formas Research Council grant 2021-01374. The
EC-Earth3 simulations and data handling were/was enabled by resources provided by the National Academic
Infrastructure for Supercomputing in Sweden (NAISS), partially funded by the Swedish Research Council
through grant agreement no. 2022-06725.








**Appendix**

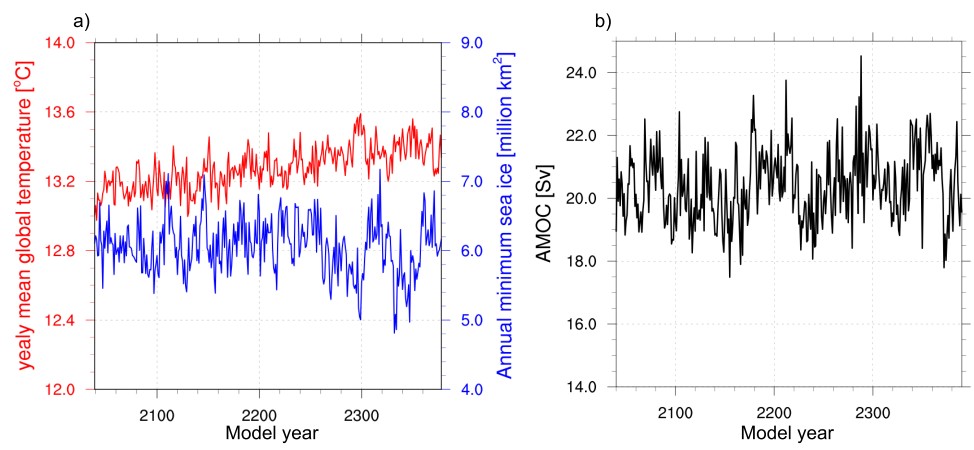

**Figure A1. Time series from the 350-year EC-Earth3-HR pre-industrial (PI) control run showing: (a) annual-mean global near-surface air temperature (°C, red) and annual-minimum sea ice area (million km², blue); (b) annual-mean AMOC strength, defined as the maximum overturning streamfunction between 20°–40°N at 800–1100 m depth.**

**Figure A2. Trend in sea ice concentration between 1980 and 2010 (% per decade). The left panel shows satellite-observed sea ice concentration, and the right panel shows EC-Earth3-HR-simulated sea ice concentration.**



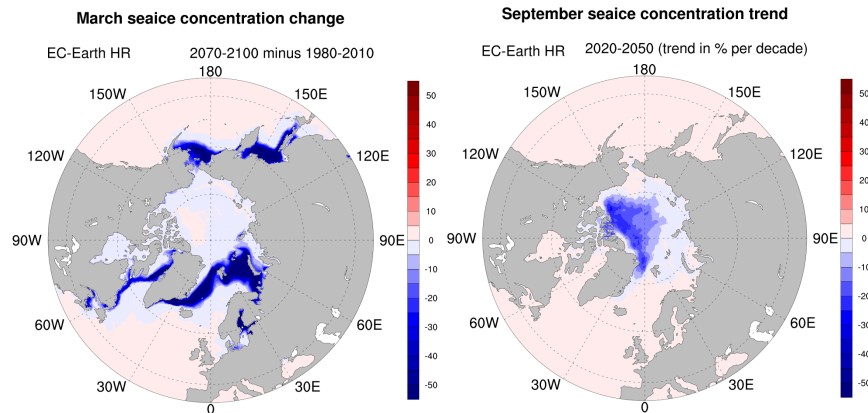


**Figure A3. The left panel shows the March sea ice concentration change (%) in EC-Earth3-HR between 1980–2010 and 2071–2100 under the SSP2-4.5 scenario, and the right panel shows the September sea ice concentration trend from 2020 to 2050 (% per decade).**





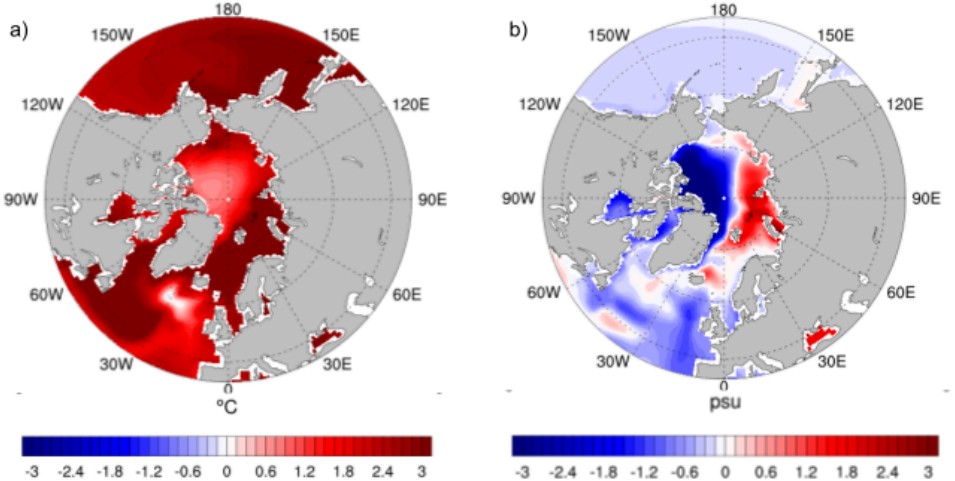


**Figure A4. Differences between the 2071–2100 and 1980–2010 periods under the SSP2-4.5 scenario for the EC-Earth3-SR ensemble mean: (a) annual mean sea surface temperature (SST, °C); (b) annual mean sea surface salinity (SSS, psu).**







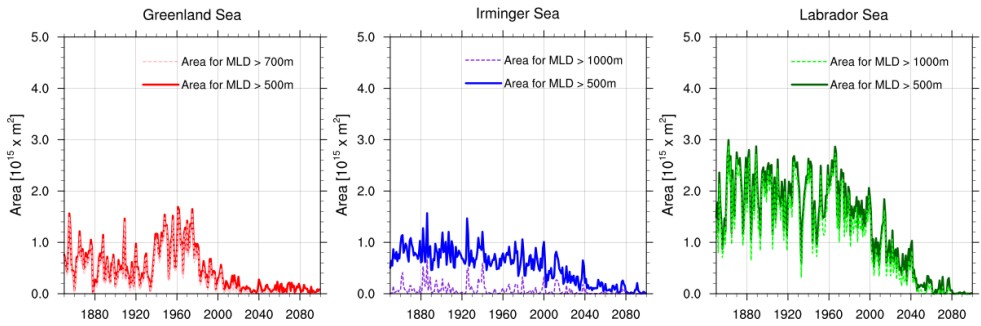

**Figure A5. Total area of water involved in deep convection, estimated by integrating the horizontal extent of the mixed layer depth (MLD) below a critical depth (1000 m: thick lines; 500 m: dashed lines) in the Greenland, Irminger, and Labrador Seas, for the historical and SSP2-4.5 simulations of EC-Earth3-HR. Area, rather than volume (cf. Figure 10), is used to facilitate comparison between the curves.**

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
