# Peer review of "Historical Climate and Future Projection in the North"

_EGUsphere, 2025_

## Referee Comment (RC2)

**Review of** ESD manuscript egusphere-2025-2653 - Historical Climate and Future Projection in the North Atlantic and Arctic: Insights from EC-Earth3 High-Resolution Simulations by KARAMI et al.

In this study, the authors discuss the performance of high-resolution climate model simulations with EC-Earth3, contrasting it to its counterpart used in CMIP6 / HighResMIP and focusing on the Arctic and North Atlantic. In the final section of the paper (Section 4.4), the authors discuss processes driving the weakening of the AMOC that is seen in this EC-Earth3 simulated under the SSP2-4.5 scenario.

**Overall impression**

The authors thoroughly analyze key features of the simulation, and how these compare to earlier versions and reanalyses. As such, it provides a benchmark for EC Earth users and is a valuable resource for those interested in these regions / this model. The manuscript is clearly written and well-structured.

However, I found section 4.4 very puzzling, i.p the calculation and physical interpretation of the deep water formation (DWF), which the authors claim is a novel way of examining the processes contributing to the AMOC strength and changes therein. In my view, that part therefore requires substantial revision. This holds for the accompanying text in the discussion too (from I.591 onwards).

**Overall assessment:**

minor revisions up to section 4.3; major revisions for section 4.4.

**Major comments**

Throughout section 4.4, the authors appear to mix up the depth (AMOC\_z) and density (AMOC\_rho) space perspective on AMOC (weakening).

In this paper, AMOC metrics are assessed in the depth perspective [AMOC\_z; Fig 8, 9a]. In addition, metrics for density changes at high latitudes are discussed [surface density and DMV; Fig 10, 11], which relate to AMOC\_rho but are discussed as if the authors expect a very tight correspondence with the AMOC\_z metrics [Fig 12b]. Finally, the authors introduce the DWF metric, which relates to AMOC\_z. I have questions about how it is defined / calculated and thus how the presented results should be interpreted.

**A - AMOC index versus DMV**

On I.449-480 the authors introduce and analyze the DMV metric. I have several questions about the chosen approach and (linked to that) the interpretation of the outcomes that I request the authors to respond to.

(1) L.455 please briefly motivate the choice of each of these depths for defining DMV.

In particular, the choice to assess DMV > 1000m for the Irminger Sea seems remarkable as that region is known to generally display shallower mixed layers than the Labrador Sea, resulting in DMV  $^{\circ}$  0 for the Irminger Sea more or less by construction [Fig 11b]. The choice of depth level thus appears to affect the conclusions: on I.461 ["consistently weak convection"] versus I. 472 "progressively shallower" [based on Fig A5b, which uses z=500m rather than 1000m to assess DMV]. To the reader, it remains unclear how the authors in the end interpret the (evolution of) DMV in the Irminger Sea

- (2) On I.464-469 the authors present correlations between the timeseries of the AMOC\_z-index and DMV, which are very high for the Labrador and Greenland Seas. While the authors seem to interpret this as a positive and encouraging outcome (I.467), for me it raises questions. It is not explicitly mentioned, but I suspect these correlations have been calculated from the curves in Fig 9a and Fig 11. If so, the high correlation values are merely a result of the fact that both curves display a dominant downward trend. The low correlation for the Irminger Sea is corroborates my suspicion, as that DMV curve has no clear trend. If the curves have not been detrended, the high correlations should not be interpreted as tight physical relations between the two variables.
  - To be honest, I would be rather worried if a model has an almost one-one-one positive correlation between variations in DMV [the volume of dense water present] and the AMOC\_z strength. DMV is the net outcome of formation (which acts to increase DMV) and export (decreases DMV). AMOC\_z, in contrast, quantifies the overall sinking that occurs north of where its index-value is taken. Deep convection represents strong vertical mixing / densification of waters and is associated with hardly any mean vertical motion where the water actually sinks is governed by vorticity dynamics (see for example Spall and Pickart 2001; Marotzke and Scott 2002; Katsman et al 2018, Bruggemann and Katsman 2019) so these are physically different things
- (3) L.468: the outcome that the strongest correlation occurs at lag zero is something that I would not expect either. While it is obvious that densification of waters at subpolar latitudes is a necessary ingredient for AMOC\_z/rho it is not sufficient: these dense waters also need to be exported southward to contribute. Since DMV is assessed at high latitudes, even if it was related one-on-one to AMOC\_z I would expect some time lag between their signals.

**B** - Definition of DWF**

On I.486-500 the authors introduce and analyze the DWF metric. I also have several questions on this.

- (1) To calculate DWF, the volume budget within a 2-layer system is considered, with the boundary between the upper and lower layers chosen at a fixed depth [I.494]
  - Why is this fixed depth now referred to as the critical depth? What is critical about it?
  - The chosen depths are "as defined above" I interpret this as the depths given on I.455 so 1000m for Irminger and Labrador Seas, 700m for Greenland Sea. Is that correct?
- (2) The volume budget for each layer is considered for the three regions defined in Fig 9b, which have some lateral boundaries in common over which exchanges are assessed. If my deduction above that depths separating upper and lower layers in the three regions differ is correct,
  - How are transports at depths 700-1000m between the Irminger and Greenland Sea handled? Do they end up in a different layer by construction? If so what does that represent physically? What impact does it have on DWF calculated in this way?
  - The sketch in Fig 12a does not reflect this difference in layer thickness between the regions; if that is what is done I think it would make sense to visualize it as well
  - If DWF is indeed calculated with differing depths for the different regions and the authors can explain and motivate this, I would like to see a discussion of how the results should be interpreted. Do the differing definitions of the 2-layer system make Fig 12b an 'apples and oranges comparison' for the DWF curves? In particular, the study by Sayol et al (2019) showed that the depth at which North Atlantic sinking peaks varies by region how do they relate their results to this?

- (3) DWF is defined as the residual of the horizontal transports into/out of a certain region. Assuming mass conservation, DWF is the net vertical transport [i.e I would define it as the regional contribution to AMOC\_z at these latitudes]. Notably, DWF is assessed at the depth separating the two layers in that region.
  - The authors seem agree that that is what they calculate [I.497] but add an interpretation [I.498] that I do not understand. Please elaborate, in particular for the Greenland Sea. In my view, water that sinks below 700m there does not automatically contribute to sustaining AMOC z as it needs to cross the Greenland-Scotland Ridge somehow for this
  - The above definition of DWF as a residual assuming mass conservation could be made more explicit
- (4) On I.515 it is stated that 'the sum of the DWF rates plus the Arctic deep flow into the Greenland Sea equals the net deep southward flow at 45N'.
  - Why is this "Arctic deep flow into the Greenland Sea" [2.2 Sv, orange arrow in Fig 12b] not automatically incorporated in the regional volume budget? [according to l.494 the lower layer reaches to the sea floor]
- (5) On I. 531-541 the authors present again correlations using timeseries with trends. The reservations expressed in remark A2 hold here as well
- (6) L.540 "critical role...": I do not think this statement ["strong relationship", "modulates"] makes sense. If calculated for a consistent depth level and for a set of regions covering all sinking regions, the AMOC\_z index and the summed DWFs should match, simply from mass conservation. Any mismatch should be attributable to transports between Atlantic and Pacific via Bering Strait (Katsman et al 2018). However, if DWF is calculated at different depths in different regions, this no longer holds as the sinking does not peak at the same depth everywhere (Sayol et al 2019)
- (7) Since DWF is assessed using a 2-layer system defined in depth space, it can provide information on the regional AMOC z contribution only [=net regional sinking].
  - The papers referred to on I. 519-520 consider AMOC\_rho, which at high latitudes, where the
    actual water mass transformation takes place, differs from AMOC\_z (see references given
    earlier). I therefore think the DWF results should not be linked to these particular studies –
    something else is assessed here.
  - I think the authors are overselling their results referring to it as giving insight in processes / drivers of AMOC weakening [I.32, I.423, I.528, I.532] in my view the study provides a regional decomposition of the (evolution of the) sinking

**References:**

- Spall, M. A., & Pickart, R. S. (2001). Where does dense water sink? A subpolar gyre example. *Journal of Physical Oceanography*, *31*(3), 810-826.
- Scott, J. R., & Marotzke, J. (2002). The location of diapycnal mixing and the meridional overturning circulation. *Journal of Physical Oceanography*, *32*(12), 3578-3595.
- Katsman, C. A., Drijfhout, S. S., Dijkstra, H. A., & Spall, M. A. (2018). Sinking of dense North Atlantic waters in a global ocean model: Location and controls. *Journal of Geophysical Research: Oceans*, 123(5), 3563-3576.
- Brüggemann, N., & Katsman, C. A. (2019). Dynamics of downwelling in an eddying marginal sea: Contrasting the Eulerian and the isopycnal perspective. *Journal of Physical Oceanography*, 49(11), 3017-3035

- Sayol, J. M., Dijkstra, H., & Katsman, C. (2019). Seasonal and regional variations of sinking in the subpolar North Atlantic from a high-resolution ocean model. *Ocean Science*, *15*(4), 1033-1053

**Minor comments / questions**

- 1. Section 1/2 several places it is stated that model results are "better" but it is not made explicit in comparison to what; some examples are I.83; I.86, I.139 [the answer is probably what is written on I.184]
- 2. Throughout the text: I think it would help the reader if the authors provide more detailed guidance on what to look focus on in the figures, pointing to explicit figure panels and or lines; some examples are I.187-196 which panel, I.303/Fig 6a which line
- 3. L. 68: 'and the processes behind the simulated weakening' what is this statement based on?
- 4. L.287 A brief summary of section 3 would be in place here
- 5. L.379: 'indicating that...' what is this statement based on?
- 6. L.471: sentence is incomplete; also I think the fact that Fig A5 shows area not volume as Fig 11 could be emphasized more
- 7. L.478 Greenland Sea convection shuts down this seems at odds with Fig 7d that shows that the surface density increases in the region where I would expect convection. Please clarify / show MLDs

**Figures**

- Figure 6: legends are very small
- Figures 7 / 9 / 10:
  - In light of what is discussed in section 4.4, I think it would be useful to add maps of MLD itself for the 2 periods in the Appendix
  - o Fig 9b: density change unclear at which depth this is
  - Why are different months/periods used for Fig 7 [=JFM], Fig 9b [=unspecified "wintertime"] and Fig 10 [DJF] ?
- Figure 10: the statement on the evolution made on I.434 may be easier to substantiate by also [potentially in the Appendix] showing rho(z,t)-rho(z,year=1850)
- Figure 11: indicate in caption that chosen depth levels for defining DMV differ [info on I.455]

**Text suggestions / typos**

- L.48 suggests
- L.54 surface temperature
- L.67 which → not sure what is refers to; I think now it is AMOC? Is that what is meant?
- L.81 verb missing in 2nd half sentence
- L.82 this sounds odd T, S profiles get better but convection properties are not OK?
- L.86 shift in response to what?
- L.96 some of these?
- L.127 reasonably reproduce
- L.243 CAA not introduced
- L.255 sentence not correct [and vague]
- L.339 very vague formulation I think this can be made more explicit
- L.362 Atl Water into the region
- L.370 SSD anomaly

- I.381 [is linked to / contributes to] and I.398 [is consistent with] it is not explicit on how strong the authors expect this link / connection between AMOC strength and MLD reduction to be
- L.393 and elsewhere: I think streamfunction is one word / no space
- L.411 seems at odds with I.394
- I.424 statement ignores the fact that formation of dense water is not sufficient for AMOC contribution the dense water also needs to flow southward
- L.426 Fig 9b shows surface density; Fig 7c shows there are large changes in that in the Arctic too; text speaks about convection / deep water formation [which is not in Fig 9b]
- L.438 retreat seems an odd word to use here
- L.445 specification that this is winter / DJF belongs right at beginning of the caption; adjust figure label/title too
- L.449 to  $\rightarrow$  for?
- L.452 sentence incomplete add 'and' before thus?
- L.467 'more closely linked' more closely than what?
- L.473 sentence not correctly formulated, please rephrase DMV based on both shallow and..?
- L.488 relative contribution to what?

---

## Author Comment (AC1)

**Reply to Review 1 of EC-Earth3-HR paper submitted to Earth system Dynamics:**

**Main comments:**

In this manuscript from Karami and colleagues a newly tuned version of EC-Eath3 at high resolution (EC-Earth3-HR) is presented. The authors compare this new model version with observations, with the ensemble of EC-EARTH3 at standard resolution, and with the previous less tuned high-resolution model performed for HighResMIP. In addition, there is a special focus on understanding AMOC changes by analysing the deep mixed layer and deep-water formation. This is a very nice set of simulations, and its description and evaluation should certainly be published. The focus on the North Atlantic, the Arctic and the AMOC allows a deeper analysis for these regions and processes which is a good idea. However, before I endorse the publication of the manuscript there are three major points that should be improved.

We sincerely thank the reviewer for their thorough and constructive comments. We provide a detailed, point-by-point response below (our responses are shown in blue). We have revised the manuscript accordingly.

First, since it is argued by the authors that the tuning is the difference with the previous high-resolution model version, I suggest that the authors explain the tuning better and reflect on it in the discussion section. For the tuning of the atmosphere, which sea surface temperature is used as boundary condition? Are the greenhouse gases set to 1850 values? What does "minimising the climate drift" mean? Which variables should not drift? How are the 5 parameters in Table 1 chosen? Are those known to be particularly important based on previous experiments with this model? How are those parameters influencing the radiative balance? Was the value of other parameters also tested?

We have expanded the description of the tuning procedure in Section 2. The revised text is included below for the reviewer's reference.

In brief, the atmospheric tuning was performed in AMIP-type simulations forced with prescribed SST and sea-ice concentrations from the PCMDI AMIP boundary condition dataset (Durack et al., 2022), using historical greenhouse gas concentrations. The term "minimising climate drift" has been removed, and the text now explicitly refers to targeting a TOA radiative imbalance close to observational estimates (Hansen et al., 2011), while avoiding physically unrealistic trends in surface air temperature and precipitation during the tuning integrations. The five atmospheric parameters listed in Table 1 were selected based on previous tuning and sensitivity experiments with EC-Earth3-SR (Döscher et al., 2022), where they were identified as particularly influential for cloud and microphysical convection processes and, consequently, the radiative balance. Due to the high computational cost of the T511 configuration, tuning was restricted to this subset of parameters.

The revised text reads as follows:

> "These AMIP tuning simulations were forced with prescribed SST and sea-ice concentrations from the PCMDI AMIP boundary condition dataset (Durack et al., 2022), and greenhouse gas concentrations followed historical values. Each tuning integration covered 20 years starting in 1990, with the final 15 years used for evaluation. The objective of the tuning was to target a

TOA radiative imbalance close to observational estimates (Hansen et al., 2011). Consistent with Döscher et al. (2022), the tuning further targeted reduced global-mean biases in the net radiative flux at the surface and in the TOA longwave flux. In addition, surface air temperature (TAS) and precipitation were monitored to avoid physically unrealistic trends during the relatively short tuning integrations. Further details of the tuning strategy and target metrics are described in Döscher et al. (2022). A major challenge was achieving a stable model state with relatively short tuning runs. To meet these radiative constraints and to reproduce the observed climate reasonably well, the model was fine-tuned by adjusting selected parameters of sub-grid-scale parameterizations. Specifically, the five atmospheric parameters listed in Table 1 were selected based on previous EC-Earth3-SR tuning and sensitivity studies (Döscher et al., 2022), where these parameters were identified as particularly influential for cloud and microphysical convection processes, which effectively control the radiation balance. Due to the very high computational cost of the T511 configuration, only this restricted subset of atmospheric parameters was retuned, while the remaining atmospheric parameters were kept at their EC-Earth3-SR default values."

Some more information is also needed for the coupled model tuning. For the choice of advection scheme, what does "overall better performance" (l.139) mean? Is it the results that are better? Were specific metrics used or did the authors look at maps of key variables to make a choice? Or was it the computational time or stability of the model that was important for this choice?

We have revised the description of the coupled-model tuning procedure by including more details about the tuning exercise and clarifying the criteria used to evaluate different tuning options, including the choice of ocean advection scheme.

The revised text reads as follows:

"Further tuning focused on the ocean component using coupled simulations. We carried out a set of 15 coupled EC-Earth3-HR simulations (50–180 years each) with different ocean and sea-ice parameter values, forced with fixed radiative conditions representative of 1981. These fixed-forcing simulations were designed to facilitate direct comparison with observations. In parallel, we conducted a coupled simulation under pre-industrial forcing conditions, in which selected tuning parameters were iteratively updated based on insights from the fixed-forcing experiments. Each new configuration was continued from the end of the previous run, under the assumption that parameter changes would induce only incremental adjustments to the model state. Except for the initial experiment initialized from Levitus climatology, all subsequent tuning experiments were concatenated. This approach limited model drift during tuning and reduced computational cost. Therefore, the concatenation of the retained tuning experiments can be regarded as a long pre-spin-up (> 200 years) for the coupled EC-Earth3-HR configuration. Following this concatenated pre-spin-up, we performed a 105-year spin-up using the final parameter set, from which a 350-year pre-industrial control simulation was initialized. Figure A1 shows that, over the final 100 years of this simulation, the global-mean TAS, SST, and Arctic sea-ice area exhibit negligible trends, indicating no ongoing drift in the simulated surface climate. The vertically averaged upper-ocean temperature (0–2000 m) and the AMOC also show negligible drift, providing no indication of ongoing drift in the deeper ocean.

The parameters tested during the tuning are listed in Table 2, with the aim of reducing model biases. The selection of adjustable parameters follows earlier EC-Earth configurations, including EC-Earth3-SR (Döscher et al., 2022) and EC-Earth3P-HR in its second configuration (Haarsma et al., 2020). The evaluation of the different tuning options was based on quantitative diagnostics and climatological bias assessments. Global-mean and map-level comparisons of key variables were used to assess the overall differences relative to observational datasets and reanalysis products. We selected the tuning choices that, overall, reduced the model biases, as discussed below.

With regards to the performance of the different advection schemes, we tested different advection schemes, including the Upstream-Biased Scheme (UBS) and, Total Variations Diminishing (TVD) approach. The UBS configuration showed a persistent warming drift in the global ocean temperature (100–1000 m) and surface heat fluxes that did not stabilise even after more than 130 years, while the TVD simulations showed a marked slowdown of drift and approached a quasi-steady state after about 60 years. The TVD scheme was therefore selected based on its long-term stability rather than on computational cost."

L.139-141: "The turbulent kinetic energy (TKE) mixing below the mixed layer was set to zero (nn_etau=0), as in EC-Earth3-SR (Doscher et al., 2022), which would otherwise lead to a significant reduction in AMOC"
Given that the AMOC changes between the time it is tunned (1850) and the time it can be compared to the RAPID array this is challenging. Was there a specific target for AMOC strength? The authors mention in the results section that the AMOC is larger in the model than in the observations. In retrospect should that value have been non zero?
The choice of nn_etau = 0 in EC-Earth3-HR follows the EC-Earth3-SR configuration, from which the high-resolution model development was initiated. In EC-Earth3-SR, setting nn_etau = 0 led to a stronger and more realistic AMOC (Döscher et al., 2022), and this motivated us to test the same choice in EC-Earth3-HR.

For the coupled tuning, our target was the observed AMOC strength of around 17–18 Sv, consistent with RAPID estimates (~17–18 Sv at 26.5°N; Smeed et al. 2018), based on the assumption that PI-to-present AMOC long-term trend is relatively small. This assumption is supported by reconstructions indicating changes of roughly 0–15% over this period (corresponding to ~0–2 Sv), as summarized in IPCC AR6 WG1. With nn_etau = 0, EC-Earth3-HR attains an AMOC slightly stronger than present observations but close to both the observational range and our pre-industrial target. We emphasize that this choice was not driven by the AMOC alone. Switching to nn_etau = 1 did not improve the overall coupled climate in EC-Earth3-HR, and sensitivity experiments with nn_etau = 0 yielded overall less bias in large-scale SST and sea-ice patterns in the North Atlantic–Arctic sector.

We have added clarifications for this choice in Section 2:
"The turbulent kinetic energy (TKE) mixing below the mixed layer was set to zero (nn_etau=0), as in EC-Earth3-SR (Döscher et al., 2022), which would otherwise lead to a significant reduction in AMOC. The tuning target was an observed AMOC strength in the range 17–18 Sv, consistent with RAPID estimates at 26.5°N (Smeed et al., 2018). It should be emphasized that this choice was not driven by the AMOC alone. Switching to nn_etau = 1

did not improve the overall coupled climate in EC-Earth3-HR, and sensitivity experiments with nn_etau = 0 produced preferable large-scale SST and sea-ice patterns in the North Atlantic–Arctic sector."

The eddy diffusivity for tracers was increased a lot compared to EC-Earth3P-HR. What was the rationale behind that choice? It is rather counter-intuitive that this parameter is now the same in the standard resolution and in the high resolution because with a higher resolution one would expect that less eddy diffusion needs to be parameterised.

We agree with the reviewer that, from a purely theoretical perspective, one would generally expect a lower lateral eddy diffusivity to be required at higher resolution. The increase of the tracer diffusivity from 300 m² s⁻¹ (EC-Earth3P-HR) to 1000 m² s⁻¹ (EC-Earth3-HR) was therefore not motivated a priori by resolution arguments, but was the outcome of targeted sensitivity experiments during the coupled tuning phase. Starting from the EC-Earth3-SR parameter set, we tested different values of rn_aht in EC-Earth3-HR in order to assess the sensitivity of the large-scale circulation and North Atlantic climate to lateral tracer mixing. Compared to a configuration with rn_aht = 300 m² s⁻¹, the configuration with rn_aht = 1000 m² s⁻¹ exhibits a stronger AMOC and deeper convection in the Labrador Sea. It also reduced the model bias for key surface climate features in the North Atlantic–Arctic sector. We therefore retain rn_aht = 1000 because of its overall better performance, while acknowledging that this improvement may partly reflect error compensation.

We have added the following sentence in Section 2 for clarity:

"Sensitivity experiments showed that rn_aht_0 = 1000 m² s⁻¹ yields a stronger AMOC and deeper Labrador Sea convection than rn_aht_0 = 300 m² s⁻¹, with an overall improved North Atlantic–Arctic surface climate."

What about the viscosity? Which type of viscosity and which coefficient is used? How does it differ between the different model versions and resolution?

We have added a description of the viscosity formulation and parameter values in Section 2, including the type of lateral momentum mixing, the reference coefficients used, and how these differ between the high-resolution and standard-resolution model configurations.

The revised manuscript text reads:

"For the lateral momentum mixing, EC-Earth-HR employs 2D-varying bilaplacian eddy viscosity with the reference value at the Equator of −6.4×10¹¹ m⁴ s⁻¹, whereas EC-Earth-SR uses a 3D-varying Laplacian eddy viscosity with the reference value of 2×10⁴ m²s⁻¹. We set the background vertical mixing parameters to 10⁻⁴ m²s⁻¹ and 10⁻⁵ m²s⁻¹ for the vertical eddy viscosity and diffusivity, respectively, compared with 1.2×10⁻⁴ m²s⁻¹ and 1.2×10⁻⁵ m²s⁻¹ in EC-Earth-SR."

The second point to improve is the connection between the main conclusions and the analysis. The abstract is written in a precise way but in the discussion and conclusion section some claims are not supported by the analysis:

l.551-552: "the updated configuration shows reduced radiative imbalances and minimal deep-ocean drift." This is not shown. I suggest adding a figure to show it. There is a clear drift of the surface

temperature in the model (Figure A1) so I expect that the deep ocean will also still be drifting.

We agree that this statement required supporting evidence. To address this, we have added two new panels to Appendix Fig. A1 from the same pre-industrial control simulation: panel (c) showing the global-mean SST and panel (d) showing the vertically averaged upper-ocean temperature (0–2000 m). While an initial adjustment is evident in the early stages of the integration, the drift in these quantities is negligible over the final 100 years of the simulation. In addition, the AMOC (panel b) exhibits no systematic drift over the full simulation, indicating no systematic drift in the deeper ocean layers. We also note that panel (a) of Figure A1 has been corrected to include the final 12 years of the simulation, which were missing in the original submission due to a plotting error.

The updated Figure A1 is shown below and has been included in the revised manuscript.

The following text is also added in Section 2, for concision and alignment, at the first mention of Figure A1:

> "Figure A1 shows that, over the final 100 years of this simulation, the global-mean TAS, SST, and Arctic sea-ice area exhibit negligible trends, indicating no ongoing drift in the simulated surface climate. The vertically averaged upper-ocean temperature (0–2000 m) and the AMOC also show negligible drift, providing no indication of ongoing drift in the deeper ocean."

Revising the sentence referred to by the reviewer to remove overstatement:

> "Compared to EC-Earth3P-HR simulations under HighResMIP, which suffered persistent biases and oceanic drift (Haarsma et al., 2020; Roberts et al., 2019), the updated configuration shows no evidence of long-term drift in the surface climate and has negligible drift in the deep ocean during the final 100 years of the pre-industrial control simulation (Figure A1)."

[Figure]

**Figure A1** *(UPDATED). Time series from the 350-year EC-Earth3-HR pre-industrial (PI) control run showing: (a) annual-mean global surface air temperature (°C, red) and September sea ice area (million km², blue); (b) annual-mean AMOC strength, defined as the maximum overturning streamfunction between 20°–40°N at 800–1100 m depth. (c) annual-mean global sea-surface temperature (SST); and (d) annual-mean global vertically averaged upper-ocean temperature (0–2000 m). An initial adjustment is evident early in the integration, while the residual drift becomes negligible over the final 100 years shown.*

The enhanced abilities or improved performance of the HR model compared to SR is not argued in the result section. Yet the following is written in the discussion and conclusion section:
l.559-561: "Time series of global and Arctic temperatures, AMOC strength, SST and sea ice area show better agreement with observations and reanalysis, indicating that higher resolution enhances the model's ability to capture transient responses and low-frequency variability"

We agree with the reviewer that clearer qualification of the HR–SR comparison was required. As stated in the manuscript, our analysis is based on a single-member simulation, which limits the robustness of comparisons between one EC-Earth3-HR realization and the ensemble mean of EC-Earth3-SR. We have therefore revised the wording of this sentence and the preceding one to better qualify the comparison and to avoid overinterpretation. The comparison is further illustrated by Figure A6, which was added to the revised manuscript in response to a subsequent reviewer comment.
The revised text now reads:
> "EC-Earth3-HR shows modest improvements in the climatological means of key variables (e.g., SST, TAS, precipitation, and sea ice) compared to EC-Earth3-SR. Time series of global and Arctic TAS, global SST, and Arctic sea ice area also show better agreement with observations and reanalysis (Figure A6)."

l.568-569: "The improved performance of the EC-Earth3-HR model in simulating Arctic–North Atlantic climate extends to key components such as Arctic sea ice, deep convection, and AMOC variability" …. l.586-587: "This is while the horizontal and vertical distribution of water masses improves in EC-Earth3-HR."

We thank the reviewer for pointing this out. Demonstrating improved performance in deep convection, AMOC variability, and the distribution of water masses would require additional targeted analyses that go beyond the scope of the present study. We have therefore removed these statements from the revised manuscript.

Based on Figure 6b it seems difficult to argue that HR performs better than LR for the Artic temperature. The improvements in AMOC trend and variability are also difficult to see Figure 9a. I suggest that the authors write more precisely what improves and what doesn't in the HR model and define clear metrics to backup claims of improvements: rate of change, standard deviation, root-mean-square-error…

We agree with the reviewer that qualitative inspection of Figures 6b and 9a alone does not allow a robust assessment of improved performance in EC-Earth3-HR. To address this, we have introduced explicit quantitative metrics, including the correlation coefficient, root-mean-square error, and mean bias relative to observations and reanalysis.
The results are summarized in the new Figure A6, which is shown below and has been added to the revised manuscript. These metrics indicate improvements in EC-Earth3-HR for global mean surface air temperature, global SST, and Arctic sea-ice area. Improvements in Arctic surface air temperature are modest, while for AMOC strength the applied metrics do not indicate improved performance in EC-Earth3-HR relative to EC-Earth3-SR (AMOC metrics are not shown). Accordingly, we have revised the manuscript text as follows:
> "Time series of global and Arctic TAS, global SST, and Arctic sea ice area also show better agreement with observations and reanalysis (Figure A6)."

The clause "indicating that higher resolution enhances the model's ability to capture transient responses and low-frequency variability" has been removed, as it is not directly supported by the applied quantitative metrics.

[Figure]

Figure A6 (Newly added). Quantitative evaluation of EC-Earth3-SR and EC-Earth3-HR performance relative to observations and reanalysis for Arctic and global surface air temperature (TAS), global sea surface temperature (SST), and Arctic sea-ice area. Shown are correlation (r), root-mean-square error (RMSE), and mean bias for each variable. Metrics are computed over the historical period: 1950–2014 for TAS, 1870–2014 for SST, and 1979–2014 for sea ice.

The third aspect to be improved is the data and code availability. This is important for other researchers to build on this work. The code used for the analysis and plot of this manuscript is not available. It is mentioned that the CDFTOOLS are used for the DWF analysis but this is not precise enough. Which script from the CDFTOOLS were used? With which options? Enough information should be provided for someone to be able to reproduce the DWF analysis. Especially since this analysis is a novelty from this manuscript.

We have made all CDFTOOLS codes used for the DWF diagnostics publicly available in this repository: https://github.com/enerle/CDF-analysis/tree/main

Specifically, the DWF analysis relies on the cdftransport operator, which was executed for each section and time slice using explicit command-line calls documented in the repository. The repository includes the exact scripts used, all options passed to cdftransport, and the post-processing steps required to compute the diagnostics. This information is sufficient to fully reproduce the DWF analysis presented in the manuscript.

The following paragraph is now added to "Calculation of Deep Water Formation (DWF) analysis" in the revised manuscript:

"In particular, volume transports across each section are calculated with the cdftransport operator applied to the EC-Earth HR output for each time slice. The exact command-line calls, including all non-default options, section definitions, and post-processing scripts used to derive the mean diagnostics, are provided in a publicly available repository (https://github.com/enerle/CDF-analysis/tree/main). This information is sufficient to reproduce the DWF analysis."

Concerning the data availability I think the following claim is misleading:

l.615-616: "Data from the EC-Earth3-HR historical and SSP2-4.5 simulations are available through any ESGF data node as part of CMIP6"

Isn't this data from the HighResMIP version of the model (called EC-Earth3P-HR in this manuscript)? How and where is the data from the simulations analysed in this paper available?

Most ESFG nodes include data of the EC-Earth3P-HR simulations, but only a few of them include the EC-Earth3-HR used in this study.

We have updated this sentence for clarity:

> "Data from the EC-Earth3-HR historical and SSP2-4.5 simulations are available through any ESGF-CoG data node (e.g. https://esg-dn1.nsc.liu.se/search/cmip6-liu/) as part of the CMIP6 project. Search for source_id="EC-Earth3-HR" and experiment_id="historical" or "ssp245". "

**Minor comments:**

l.148: "the freshwater correction value was slightly modified to oas_mb_fluxcorr=1.07945"

Modified compared to what? The value is the same in the two models compared in Table 2. Also, a short explanation of what this parameter is would be useful. The name "oas_mb_fluxcorr" is not used in Doscher et al. (2022). Is this needed because IFS does not conserve water? Heat is also probably not conserved in IFS, is it also corrected for?

We did not change this parameter; the sentence and table have been corrected accordingly. oas_mb_fluxcorr is indeed needed to counteract the P-E imbalance of IFS.

l.160-162: How long is this "long concatenated pre-spin-up run"?

It spans more than 200 years, which has been added to the revised text.

Fig. 1a: Could you compare with longer reconstructions of TAS? For example, ERA5 is now available from 1940 while here only data from around 1980 is used.

Updated figure is added to the revised version.

l.243: "CAA" is not defined.

Corrected to Canadian Arctic Archipelago

l.321: "PrI" is not defined

Corrected to pre-industrial

Fig. 9b: Is that the density change at the surface? It would be good to mention it in the caption.

Added

l.412: It is not clear what "potential" means here. I would understand if "potential" was used for a claim about the real world but here it is about models.

We removed "potential" from the sentence.

l.464 and l.533: These correlation coefficients need more context. Are they computed on the yearly averaged data? Are the time series detrended? How? The trend is not linear. Are the correlations reflecting the co-variability at inter-annual time scale or a trend common to the time series?

To assess how strongly the long-term AMOC weakening co-evolves with regional DMV and DWF changes, we computed the correlations from the full, non-detrended time series. Our focus was therefore on trend-related co-variation rather than detrended variability. While these correlations largely reflect shared trends, they do not, by themselves, demonstrate physical coupling. To avoid overinterpretation, we removed the DMV correlations and only retained the DWF analysis in the revision. Our aim was not to imply a tight dynamical link, but rather to examine whether DWF diagnostics evolve consistently with the AMOC index under anthropogenic forcing.

Citation: https://doi.org/10.5194/egusphere-2025-2653-RC1

**References**

Döscher, R., Acosta, M., Alessandri, A., et al. (2022). The EC-Earth3 earth system model for the climate model intercomparison project 6. Geoscientific Model Development, 15, 2973–3020. https://doi.org/10.5194/gmd-15-2973-2022

Durack, Paul J.; Taylor, Karl E.; Ames, Sasha; Po-Chedley, Stephen; Mauzey, Christopher (2022). PCMDI AMIP SST and sea-ice boundary conditions version 1.1.8.Earth System Grid Federation. https://doi.org/10.22033/ESGF/input4MIPs.16921

Hansen, J., Sato, M., Kharecha, P., and von Schuckmann, K.: Earth's energy imbalance and implications, Atmos. Chem. Phys., 11, 13421–13449, https://doi.org/10.5194/acp-11-13421-2011, 2011.

---

## Author Comment (AC2)

**Review of** ESD manuscript egusphere-2025-2653 **-** Historical Climate and Future Projection in the North Atlantic and Arctic: Insights from EC-Earth3 High-Resolution Simulations by KARAMI et al.

In this study, the authors discuss the performance of high-resolution climate model simulations with EC-Earth3, contrasting it to its counterpart used in CMIP6 / HighResMIP and focusing on the Arctic and North Atlantic. In the final section of the paper (Section 4.4), the authors discuss processes driving the weakening of the AMOC that is seen in this EC-Earth3 simulated under the SSP2-4.5 scenario.

**Overall impression**
The authors thoroughly analyze key features of the simulation, and how these compare to earlier versions and reanalyses. As such, it provides a benchmark for EC Earth users and is a valuable resource for those interested in these regions / this model. The manuscript is clearly written and well-structured.
However, I found section 4.4 very puzzling, i.p the calculation and physical interpretation of the deep water formation (DWF), which the authors claim is a novel way of examining the processes contributing to the AMOC strength and changes therein. In my view, that part therefore requires substantial revision. This holds for the accompanying text in the discussion too (from l.591 onwards).

**Overall assessment:**
minor revisions up to section 4.3; major revisions for section 4.4.
We sincerely thank the reviewer for her constructive and detailed comments. Our point-by-point response is provided below (our text is in blue). Where appropriate, we quote revised text directly; otherwise, we describe the changes that will be implemented in the revised manuscript. The revised DWF calculation using a uniform 1000 m reference depth has been implemented for the response and is illustrated in the response figures.

Major comments
Throughout section 4.4, the authors appear to mix up the depth (AMOC_z) and density (AMOC_rho) space perspective on AMOC (weakening).
In this paper, AMOC metrics are assessed in the depth perspective [AMOC_z; Fig 8, 9a]. In addition, metrics for density changes at high latitudes are discussed [surface density and DMV; Fig 10, 11], which relate to AMOC_rho but are discussed as if the authors expect a very tight correspondence with the AMOC_z metrics [Fig 12b]. Finally, the authors introduce the DWF metric, which relates to AMOC_z. I have questions about how it is defined / calculated and thus how the presented results should be interpreted.
Our analysis and all AMOC diagnostics presented in the manuscript are defined and interpreted in z-space and we do not intend to conflate AMOC_z and AMOC_ρ metrics. However, we acknowledge that parts of the methodological description and the physical interpretation in section 4.4 may not have been sufficiently explicit, which could give the impression of an assumed one-to-one correspondence. We clarify these aspects and the definition and interpretation of the DWF metric in detail in our point-by-point responses below.

**A - AMOC index versus DMV**
On l.449-480 the authors introduce and analyze the DMV metric. I have several questions about the chosen approach and (linked to that) the interpretation of the outcomes that I

request the authors  to respond to.

(1) L.455 please briefly motivate the choice of each of these depths for defining DMV.

In particular, the choice to assess DMV > 1000m for the Irminger Sea seems remarkable as that region is known to generally display shallower mixed layers than the Labrador Sea, resulting in DMV ~ 0 for the Irminger Sea more or less by construction [Fig 11b]. The choice of depth level thus appears to affect the conclusions: on l.461 ["consistently weak convection"] versus l. 472 "progressively shallower" [based on Fig A5b, which uses z=500m rather than 1000m to assess DMV]. To the reader, it remains unclear how the authors in the end interpret the (evolution of) DMV in the Irminger Sea

We chose to follow the definition of deep convection adopted by Brodeau & Koenigk (2016), who themselves build on Marshall & Schott (1999). These studies emphasise that there is no universal depth criterion and that the appropriate threshold depends on the renewal depth of the water masses in each basin. For the Labrador Sea, observations support a threshold near 1000 m, while in the Nordic Seas, the relevant depth is determined by the Denmark Strait overflow sill, which motivates the use of 700 m. For the Irminger Sea, however, there is no consensus. While observations show mixed layers that are generally shallower than those in the Labrador Sea, deep events exceeding 1000 m have been reported (Våge et al., 2009; Piron et al., 2017). Furthermore, our EC-Earth3-HR simulation produces mixed layers that are deeper than 1000 m during the historical period, albeit over a limited area (response Figure 1). Using 1000 m does not yield a DMV of zero, but rather isolates only the deepest Irminger events and thus results in weaker values compared with those in the Labrador and Greenland Seas. In the absence of an established, basin-specific threshold, we therefore choose 1000 m as our primary choice.

Clarifications regarding the choice of reference depths used to define DMV have been added to the DMV section of the revised manuscript, as shown below:
"To compute the DMV, a reference depth must be specified to define deep mixed layers. We follow the definition of deep convection adopted by Brodeau and Koenigk (2016), building on Marshall and Schott (1999), which emphasises that the depth threshold should reflect the renewal depth of water masses in each basin. Accordingly, Brodeau and Koenigk (2016) used an observation-based threshold of 1000 m for the Labrador Sea and a shallower threshold of 700 m in the Nordic Seas, reflecting the depth of the Denmark Strait overflow sill. For the Irminger Sea, however, no consensus threshold exists; while mixed layers are generally shallower than 1000 m, deep convection events exceeding this depth have been reported (Våge et al., 2009; Piron et al., 2017) and are also reproduced in our EC-Earth3-HR simulation. In the absence of a basin-specific criterion, we therefore adopt 1000 m as the reference depth for diagnosing deep convection in the Irminger Sea."

To address the reviewer's concern regarding interpretation, we additionally assessed a shallower threshold (500 m), as already shown in Figure A5 of the submitted manuscript. While the 1000 m threshold isolates only the deepest Irminger events, the 500 m threshold is used to diagnose the evolution of less deep events. We now removed the mentioned sentence as it was not clear and revised the explanation as

following:

"Sensitivity analyses using different critical depth thresholds were also performed with a shallower criterion (500 m; Fig. A5), and the results confirm the robustness of those obtained using the 1000 m criterion discussed above."

**MLD[m] in JFM**

averaged over 1980-2010    m

[Figure]

**Response Figure 1. EC-Earth3-HR winter mean (JFM) mixed layer depth (MLD, m) averaged over 1980-2010.**

(2) On l.464-469 the authors present correlations between the timeseries of the AMOC_z-index and DMV, which are very high for the Labrador and Greenland Seas. While the authors seem to interpret this as a positive and encouraging outcome (l.467), for me it raises questions. It is not explicitly mentioned, but I suspect these correlations have been calculated from the curves in Fig 9a and Fig 11. If so, the high correlation values are merely a result of the fact that both curves display a dominant downward trend. The low correlation for the Irminger Sea is corroborates my suspicion, as that DMV curve has no clear trend. If the curves have not been detrended, the high correlations should not be interpreted as tight physical relations between the two variables.

The correlations were indeed computed from the full, non-detrended time series and were intended to quantify how strongly the long-term AMOC_z weakening is statistically associated with regional DMV changes. In that sense, our focus was on the trend-related co-evolution rather than on variability after detrending. However, we agree that these correlations mainly reflect co-trending and do not demonstrate a tight physical coupling, and they are not essential for our main conclusions. To avoid potential overinterpretation, we have therefore removed this correlation analysis for DMV in the revised manuscript.

To be honest, I would be rather worried if a model has an almost one-one-one positive correlation between variations in DMV [the volume of dense water present] and the AMOC_z strength. DMV is the net outcome of formation (which acts to increase DMV) and export (decreases DMV). AMOC_z, in contrast, quantifies the overall sinking that occurs north of where its index-value is taken. Deep convection represents strong vertical mixing / densification of waters and is associated with hardly any mean vertical motion – where the water actually sinks is governed by vorticity dynamics (see for example Spall and Pickart 2001; Marotzke and Scott 2002; Katsman et al 2018, Bruggemann and Katsman 2019) so these are physically different things

We thank the reviewer for this very helpful physical clarification and for pointing us to the relevant literature. We fully agree that DMV and AMOC_z quantify fundamentally different physical processes: DMV reflects the net balance between dense water formation and export, whereas AMOC_z represents the integrated overturning and is not directly tied to local vertical sinking associated with deep convection. We also agree that a near one-to-one correspondence between the two would indeed be physically questionable. Our original intention was not to imply such a direct mechanistic coupling, but rather to explore their statistical co-evolution under long-term forcing. In line with our response above, this correlation analysis is therefore no longer included in the revised manuscript to avoid any misleading physical interpretation.

(3) L.468: the outcome that the strongest correlation occurs at lag zero is something that I would not expect either. While it is obvious that densification of waters at subpolar latitudes is a necessary ingredient for AMOC_z/rho it is not sufficient: these dense waters also need to be exported southward to contribute. Since DMV is assessed at high latitudes, even if it was related one-on-one to AMOC_z I would expect some time lag between their signals.

We agree that, from a dynamical perspective, a time lag between high-latitude densification and the AMOC response is generally expected, since dense waters must first be exported southward to contribute to the overturning circulation. The absence of a clear lag in our original analysis could arise from the fact that the correlations were computed using the full, non-detrended time series and are therefore dominated by the common forced trend rather than by propagated variability. When focusing on detrended signals or non-transient simulations, a lead–lag relationship between DMV and AMOC does emerge, consistent with previous findings (Fig. 14 in Brodeau and Koenigk, 2016). In line with our response above, we therefore no longer include this correlation analysis in the revised manuscript to avoid potential misinterpretation.

**B - Definition of DWF**

On l.486-500 the authors introduce and analyze the DWF metric. I also have several questions on this.

(1) To calculate DWF, the volume budget within a 2-layer system is considered, with the boundary between the upper and lower layers chosen at a fixed depth [l.494]
- Why is this fixed depth now referred to as the critical depth? What is critical about it?
- The chosen depths are "as defined above" – I interpret this as the depths given on l.455 so 1000m for Irminger and Labrador Seas, 700m for Greenland Sea. Is that correct?

In the original manuscript, the term *critical depth* was adopted from from Brodeau and Koenigk (2016); however, it was not intended to imply a dynamical threshold and could therefore be misleading in this context. In the revised version, we no longer use this terminology and instead refer simply to a "reference depth".

The reviewer is also correct that the reference depths in the original submission were 1000 m for the Labrador and Irminger Seas and 700 m for the Greenland Sea, following the choices used for the DMV metric. As explained in our reply to comment (2) below, we now adopt a single, uniform reference depth of 1000 m for all regions to ensure a consistent depth-space framework.

(2) The volume budget for each layer is considered for the three regions defined in Fig 9b, which have some lateral boundaries in common over which exchanges are assessed. If my deduction above that depths separating upper and lower layers in the three regions differ is correct, - How are transports at depths 700-1000m between the Irminger and Greenland Sea handled? Do they end up in a different layer by construction? If so what does that represent physically? What impact does it have on DWF calculated in this way?

- The sketch in Fig 12a does not reflect this difference in layer thickness between the regions; if that is what is done I think it would make sense to visualize it as well
- If DWF is indeed calculated with differing depths for the different regions and the authors can explain and motivate this, I would like to see a discussion of how the results should be interpreted. Do the differing definitions of the 2-layer system make Fig 12b an 'apples and oranges comparison' for the DWF curves? In particular, the study by Sayol et al (2019)

showed that the depth at which North Atlantic sinking peaks varies by region – how do they relate their results to this?

To avoid any misunderstanding, we clarify that the two-layer framework is used to diagnose regional volume budgets across a fixed reference depth, based on transports evaluated at the gateway or boundary sections. The two-layer split is therefore not meant to imply an explicit vertical structure within the interior of the boxes, but provides a consistent depth-space framework for closing the volume budget using boundary exchanges. Transports at shared boundaries (e.g. between the Irminger and Greenland Seas) are always evaluated using the same reference depth at a given section and are therefore identical.

In this study, we construct a volume-budget diagnostic formulated in depth space. For each region, we ask the following question: given the horizontal inflow above a chosen reference depth and the horizontal outflow below that depth at the region's boundaries, what net downward transfer across that depth is required by volume conservation within the region? This residual defines our depth-space deep water formation (DWF) index. This diagnostic does not imply that sinking is necessarily maximised at the reference depth; rather, it provides a budget-consistent measure of the net downward volume transfer across that depth surface.

We agree with the reviewer that the interpretation of DWF index becomes ambiguous if different reference depths are used for different regions, as a two-layer volume budget only closes cleanly if all boundary exchanges for a given region are expressed relative to the same interface depth. To remove this ambiguity and to ensure direct comparability across regions, **we therefore adopt a single, uniform reference depth of 1000 m for all gateway and section transports in the revised manuscript**. With this choice, all transports are consistently partitioned into 0–1000 m and 1000 m–bottom contributions. This brings the method into closer alignment with common depth-space formulations of the overturning circulation, for which transports near 1000 m are often used to characterise the depth of the AMOC_z maximum. It should be noted that if a boundary section is shallower than 1000 m, transport below 1000 m is zero by geometry, and contribution is only to the net top flow.

Additionally we performed sensitivity tests using several alternative uniform reference depths (**Response Figure 2**). These tests show that the depth at which the DWF index

attains its maximum varies by region—around 600 m in the GIN Sea, near 1000 m in the Irminger Sea, and closer to 1500 m in the Labrador Sea, and remain relatively stable over time, with only a slight upward shift in the Labrador Sea. This is consistent with Sayol et al. (2019), who demonstrated that the depth of maximum sinking varies regionally in the North Atlantic. However, as our primary aim is to decompose the AMOC_z volume budget rather than to identify the precise depth of water-mass transformation in each basin, we retain 1000 m as a pragmatic, common reference depth for the main analysis. Under this choice, the deviation from the local maximum DWF is small in the Labrador Sea but larger in the GIN Sea.

**Importantly, the overall conclusions drawn from Figure 12 of the submitted manuscript are unaffected by this choice and remain robust** under the revised formulation using a uniform reference depth of 1000 m (Response Figure 3). While the magnitude of the DWF index increases slightly in the Irminger Sea and decreases in the GIN Sea when adopting a uniform 1000 m reference depth, the projected future changes are much less sensitive to the choice of reference depth. Response Figure 2 further confirms that the projected reduction in DWF toward the future is robust across different choices of reference depth. The consistent temporal evolution across reference depths indicates that the signals discussed in this section reflect changes in the overturning-related volume budget rather than artefacts of the layer definition.

These clarifications regarding the DWF methodology, the choice of a uniform 1000 m reference depth, and the resulting updated values will be incorporated into the revised manuscript. Figure 12b is already updated, and Figure 12a will be updated accordingly. We will also document the associated depth sensitivity, as shown in Response Fig. 2, in the supplementary material.

[Figure]

**Response Figure 2. Sensitivity of the deep-water formation (DWF) index to the choice of reference depth for the Labrador (LAB), Irminger (IRM), and GIN seas, shown for the historical period (mean over 1980–2014) and two future periods (mean over 2015–2040 and 2070–2100). While the magnitude of the DWF index depends on the reference depth, the relative regional contrasts and projected future changes are robust.**

[Figure]

**Response Figure 3.** Updated version of Fig. 12b in the original manuscript; in this figure, all transports and DWFs are computed using a uniform reference depth of 1000 m. Time series of southward deep transport at 45°N (black) and deep water formation (DWF) rates in the Irminger (blue), Labrador (green), and Greenland (red) seas, Arctic flow below 1000m (light blue), together with sum of all DWFs ( DWF SUM in grey) and DWF SUM plus deep Arctic flow (pink dashed), from 1850 to 2100 in EC-Earth3-HR. The close agreement between the southward transport at 45°N with DWF SUM plus Arctic flow demonstrates volume conservation of the system. All values are in Sverdrups (Sv).

(3) DWF is defined as the residual of the horizontal transports into/out of a certain region. Assuming  mass conservation, DWF is the net vertical transport [i.e I would define it as the regional  contribution to AMOC_z at these latitudes]. Notably, DWF is assessed at the depth separating  the two layers in that region.
- The authors seem agree that that is what they calculate [l.497] but add an interpretation  [l.498] that I do not understand. Please elaborate, in particular for the Greenland Sea. In my   view, water that sinks below 700m there does not automatically contribute to sustaining   AMOC_z as it needs to cross the Greenland-Scotland Ridge somehow for this
- The above definition of DWF as a residual assuming mass conservation could be made more  explicit

We thank the reviewer for helping us to refine our terminology. As clarified in our response to comment (2), the DWF diagnostic used in this study represents the net vertical transport across a fixed reference depth implied by the horizontal volume transports into and out of each region. In other words, it is the residual of the regional horizontal volume budget and corresponds to that region's contribution to the overturning circulation *in depth space* (AMOC_z) at that latitude.

We have revised the sentence as follows:
> "This pattern indicates a net vertical transport across the reference depth, which we interpret as a diagnostic of regional contributions to the overturning circulation."

The reviewer is also correct that our original phrasing ("feeds into and sustains the AMOC") was too strong. The diagnostic quantifies **downward volume convergence across the reference depth**, not diapycnal water-mass transformation or the physical locations where dense waters sink. As the reviewer notes, in the Greenland Sea only a fraction of the water is exported across the Greenland–Scotland Ridge; the remainder may recirculate or be influenced by Arctic inflows. The DWF index should therefore be interpreted as a **regional decomposition of the AMOC_z volume budget**, rather than as implying that all locally diagnosed downward transfer contributes directly to the southward deep limb.

To avoid misunderstanding, the corresponding sentence in the manuscript has been revised to:
> "This pattern indicates a net vertical transport across the reference depth, which we interpret as a diagnostic of regional contributions to the overturning circulation."

(4) On l.515 it is stated that 'the sum of the DWF rates plus the Arctic deep flow into the Greenland  Sea equals the net deep southward flow at 45N'.
   - Why is this "Arctic deep flow into the Greenland Sea" [2.2 Sv, orange arrow in Fig 12b] not  automatically incorporated in the regional volume budget? [according to l.494 the lower  layer reaches to the sea floor]

When the three subpolar regions (Labrador, Irminger and Greenland Seas) are considered collectively, they constitute a larger control volume whose southern boundary is defined by the section at 45°N, and its northern boundaries are the Arctic gateways (Fram Strait, the Barents Sea opening, and the northern boundary of our Labrador box). By volume conservation, the deep southward transport across 45°N (black line in response Figure 3) must balance:
   1. sum of our DWF indices (DWF SUM; grey line in Response Figure 3) and
   2. net deep inflow from the Arctic below 1000m (light blue line in Response Figure 3)
In descriptive form, this relationship can be written as:
**deep transport at 45°N = DWF in Labrador + DWF in Irminger + DWF in Greenland + deep net flow (below 1000 m) from the Arctic.** To demonstrate this, we refer to the Response Figure 3.

In this framework, the Arctic deep flow closes the volume budget of the three-box system, and it therefore illustrated as a distinct lower-layer source in the combined budget. Fram Strait is the main gateway that provides appreciable exchange below 1000 m, while Denmark Strait and the Barents Sea Opening are shallower and contribute exclusively to the upper-layer budget.

We will clarify this point in the revised manuscript by explicitly including the relevant transports, as shown in Response Figure 3, and by adding the following text: "This balance follows from volume conservation applied to the combined subpolar control volume, whose southern boundary is at 45°N and whose northern boundaries are the Arctic gateways."

(5) On l. 531-541 the authors present again correlations using timeseries with trends. The  reservations expressed in remark A2 hold here as well

As with the DMV correlations discussed in Comment A2, the correlations between DWF and the AMOC index in the original manuscript were computed from the full (non-detrended) time series and therefore primarily reflect the common long-term forced trend. Our intention was not to infer a tight dynamical coupling, but to assess whether the depth-space volume-budget diagnostic (summed regional DWF indices) evolves consistently with the AMOC_z index under anthropogenic forcing. We note that this point is already stated in the original manuscript, where we report the substantially weaker correlations over 1850–1980 to illustrate that the strong values over 1850–2100 are dominated by the forced trend.

To avoid any misunderstanding, we revised the sentence at line 531 as following:
> "In addition to these mean changes, the temporal covariability between DWF and the AMOC index is examined."

(6) L.540 "critical role…": I do not think this statement ["strong relationship", "modulates"] makes sense. If calculated for a consistent depth level and for a set of regions covering all sinking regions, the AMOC_z index and the summed DWFs should match, simply from mass conservation. Any mismatch should be attributable to transports between Atlantic and Pacific via Bering Strait (Katsman et al 2018). However, if DWF is calculated at different depths in different regions, this no longer holds as the sinking does not peak at the same depth everywhere (Sayol et al 2019)

We hope that the clarifications provided in our responses above address the reviewer's concerns and make clear that, within our framework, volume conservation is maintained even when different reference depths are used for the individual basins. Nevertheless, as noted in our responses to comments (1) and (2), we now adopt a uniform reference depth of 1000 m across all regions.

The reviewer is correct that this statement was an overinterpretation based solely on a statistical correlation. We, therefore, removed this sentence.

(7) Since DWF is assessed using a 2-layer system defined in depth space, it can provide information on the regional AMOC_z contribution only [=net regional sinking].
  - The papers referred to on l. 519-520 consider AMOC_rho, which at high latitudes, where the actual water mass transformation takes place, differs from AMOC_z (see references given earlier). I therefore think the DWF results should not be linked to these particular studies – something else is assessed here.
  - I think the authors are overselling their results referring to it as giving insight in processes / drivers of AMOC weakening [l.32, l.423, l.528, l.532 ] – in my view the study provides a regional decomposition of the (evolution of the) sinking

We agree that our diagnostic, being formulated entirely in depth space, quantifies **regional contributions to AMOC_z** and does not capture water-mass transformation pathways or isopycnal overturning as discussed in AMOC_ρ studies. The reviewer is therefore correct that we should be careful when comparing to other studies based on density-space diagnostics.

With regards to the second point, we agree that our analysis is primarily diagnostic and provides a regional, depth-space decomposition of AMOC_z rather than isolating individual

physical mechanisms in a causal sense. To avoid overstatement, we will revise the text at l.32, l.423, l.528, and l.532 to replace references to "drivers" or "processes" with more precise language emphasising regional decomposition and volume-budget diagnostics. We note, however, that the framework does provide insight into how individual subpolar basins and the Arctic contribution project onto the large-scale overturning, thereby clarifying the structural origins of the simulated AMOC weakening.

References:

- Spall, M. A., & Pickart, R. S. (2001). Where does dense water sink? A subpolar gyre example. *Journal of Physical Oceanography*, *31*(3), 810-826.
- Scott, J. R., & Marotzke, J. (2002). The location of diapycnal mixing and the meridional overturning circulation. *Journal of Physical Oceanography*, *32*(12), 3578-3595. - Katsman, C. A., Drijfhout, S. S., Dijkstra, H. A., & Spall, M. A. (2018). Sinking of dense North Atlantic waters in a global ocean model: Location and controls. *Journal of Geophysical Research: Oceans*, *123*(5), 3563-3576.
- Brüggemann, N., & Katsman, C. A. (2019). Dynamics of downwelling in an eddying marginal sea: Contrasting the Eulerian and the isopycnal perspective. *Journal of Physical Oceanography*, *49*(11), 3017-3035
- Sayol, J. M., Dijkstra, H., & Katsman, C. (2019). Seasonal and regional variations of sinking in the subpolar North Atlantic from a high-resolution ocean model. *Ocean Science*, *15*(4), 1033- 1053

Minor comments / questions
1. Section 1/2 – several places it is stated that model results are "better" but it is not made explicit in comparison to what; some examples are l.83; l.86, l.139 [the answer is probably what is written on l.184]
We have improved these in the revised version.

2. Throughout the text: I think it would help the reader if the authors provide more detailed guidance on what to look focus on in the figures, pointing to explicit figure panels and or lines; some examples are l.187-196 – which panel, l.303/Fig 6a - which line
We will consider this in the revised version.

3. L. 68: 'and the processes behind the simulated weakening' – what is this statement based on?
If this was meant for line 64, we will try to be specific or remove the statement.

4. L.287 A brief summary of section 3 would be in place here
Brief summary is included in the Discussion.

5. L.379: 'indicating that…' – what is this statement based on?
This is explained in the sentence before.

6. L.471: sentence is incomplete; also I think the fact that Fig A5 shows area not volume as Fig 11 could be emphasized more
We were unable to identify the incomplete sentence referred to in line 471 and would

appreciate clarification if possible. Regarding Fig. A5, the figure intentionally shows area rather than volume, as this representation brings the different regions onto a more comparable range on the y-axis and facilitates visual comparison between the boxes.

7. L.478 Greenland Sea convection shuts down – this seems at odds with Fig 7d that shows that the surface density increases in the region where I would expect convection. Please clarify / show MLDs

In Response Figure 1, we show the MLD. While there are indeed regions in the Greenland Sea where the MLD increases, this is on the order of 200–300 m. These depths are shallower than those required for deep or dense-mode convection and therefore do not contribute to DMV used in our analysis.

Figures
- Figure 6: legends are very small
Will be adjusted
- Figures 7 / 9 / 10:
       o In light of what is discussed in section 4.4, I think it would be useful to add maps of MLD itself for the 2 periods in the Appendix
       Yes, we can include the figure like Response Figure 1 in the revised version
       o Fig 9b: density change - unclear at which depth this is
       At the surface
       o Why are different months/periods used for Fig 7 [=JFM], Fig 9b [=unspecified "wintertime"] and Fig 10 [DJF] ?
       Same JFM was used for all and Figure 10 caption will be corrected.
- Figure 10: the statement on the evolution made on l.434 may be easier to substantiate by also [potentially in the Appendix] showing rho(z,t)-rho(z,year=1850)
While this would be a useful additional diagnostic, including all such analyses would substantially expand the number of figures without changing the conclusions. We therefore do not add this figure, but try to better clarify the text around line 434.
- Figure 11: indicate in caption that chosen depth levels for defining DMV differ [info on l.455]
Will be added in the revised text.

Text suggestions / typos
- L.48 suggests
- L.54 surface temperature
- L.67 which ☐ not sure what is refers to; I think now it is AMOC? Is that what is meant? - L.81 verb missing in 2nd half sentence
- L.82 this sounds odd – T, S profiles get better but convection properties are not OK? - L.86 shift in response to what?
- L.96 some of these?
- L.127 reasonably reproduce
- L.243 CAA not introduced
- L.255 sentence not correct [and vague]
- L.339 very vague formulation – I think this can be made more explicit
- L.362 Atl Water into the region
- L.370 SSD anomaly
- l.381 [is linked to / contributes to] and l.398 [is consistent with] it is not explicit on how

strong  the authors expect this link / connection between AMOC strength and MLD reduction to be  - L.393 and elsewhere: I think streamfunction is one word / no space
- L.411 seems at odds with l.394
- l.424 statement ignores the fact that formation of dense water is not sufficient for AMOC  contribution – the dense water also needs to flow southward
- L.426 Fig 9b shows surface density; Fig 7c shows there are large changes in that in the Arctic too;  text speaks about convection / deep water formation [which is not in Fig 9b] - L.438 retreat seems an odd word to use here
- L.445 specification that this is winter / DJF belongs right at beginning of the caption; adjust figure label/title too
- L.449 to □ for?
- L.452 sentence incomplete – add 'and' before thus?
- L.467 'more closely linked' – more closely than what?
- L.473 sentence not correctly formulated, please rephrase – DMV based on both shallow and..?  - L.488 relative contribution to what?
All typographical errors and minor textual corrections noted in this comment will be incorporated in the revised manuscript.

**References**

Brodeau, L., & Koenigk, T. (2016). Extinction of the northern oceanic deep convection in an ensemble of climate model simulations of the 20th and 21st centuries. Climate Dynamics, 46, 1399-1415. https://doi.org/10.1007/s00382-015-2736-5

Marshall J, Schott F (1999) Open-ocean convection: observations, theory, and models. Rev Geophys 37:1–64. doi:10.1029/98RG02739

Våge, K., Pickart, R., Thierry, V. et al. Surprising return of deep convection to the subpolar North Atlantic Ocean in winter 2007–2008. Nature Geosci 2, 67–72 (2009). https://doi.org/10.1038/ngeo382

Piron, A., V. Thierry, H. Mercier, and G. Caniaux (2017), Gyre-scale deep convection in the subpolar North Atlantic Ocean during winter 2014–2015, Geophys. Res. Lett., 44, 1439–1447, doi:10.1002/2016GL071895.